# High temperature and humidity in the environment disrupt bile acid metabolism, the gut microbiome, and GLP-1 secretion in mice
Song Chen[1,12], Zongren Hu[2,12], Jianbang Tang[3,12], Haipeng Zhu[4], Yuhua Zheng[5], Jiedong Xiao[5], Youhua Xu[6], Yao Wang [5], Yi Luo[5], Xiaoying Mo[5], Yalan Wu [5], Jianwen Guo [7,8,9] ✉, Yongliang Zhang [10,11] ✉ & Huanhuan Luo [5,7] ✉

High temperature and humidity in the environment are known to be associated with discomfort and disease, yet the underlying mechanisms remain unclear. We observed a decrease in plasma glucagon-like peptide-1 levels in response to high-temperature and humidity conditions. Through 16S rRNA gene sequencing, alterations in the gut microbiota composition were identified following exposure to high temperature and humidity conditions. Notably, changes in the gut microbiota have been implicated in bile acid synthesis. Further analysis revealed a decrease in lithocholic acid levels in high-temperature and humidity conditions. Subsequent in vitro experiments demonstrated that lithocholic acid increases glucagon-like peptide-1 secretion in NCI-H716 cells. Proteomic analysis indicated upregulation of farnesoid X receptor expression in the ileum. In vitro experiments revealed that the combination of lithocholic acid with farnesoid X receptor inhibitors resulted in a significant increase in GLP-1 levels compared to lithocholic acid alone. In this study, we elucidate the mechanism by which reduced lithocholic acid suppresses glucagon-like peptide 1 via farnesoid X receptor activation under high-temperature and humidity condition.

Recent research has demonstrated the profound impact of climate conditions, particularly temperature and humidity, on human health. Notably, cold acclimation has been found to enhance peripheral insulin sensitivity in individuals with type 2 diabetes[1]. This effect may be explained by alterations to the gut microbiota, which can lead to increased energy expenditure[2]. Specifically, exposure to cold temperatures stimulates a metabolic shift, thereby coordinating cholesterol conversion to bile acids in the liver via an alternative synthesis pathway. This shift results in increased plasma bile acid levels and distinct changes in the gut microbiota composition, accompanied by increased heat production[3].

However, the effects of high temperatures, as opposed to cold, on the gut microbiota and metabolic processes, remain less understood. Similarly, atmospheric humidity is known to contribute to physiological disturbances and diseases[4]. For instance, high relative humidity has been associated with increased diabetes morbidity among elderly individuals[5].

[1]Science and Technology Innovation Center, Guangzhou University of Chinese Medicine, Guangzhou, China. [2]Department of Rehabilitation and Healthcare, Hunan University of Medicine, Huaihua, China. [3]Zhongshan Hospital of Traditional Chinese Medicine Affiliated to Guangzhou University of Chinese Medicine, Zhongshan, China. [4]Dongguan People's hospital, Dongguan, China. [5]School of Basic Medicine, Guangzhou University of Chinese Medicine, Guangzhou, China. [6]Faculty of Chinese Medicine, State Key Laboratory of Quality Research in Chinese Medicines, Macau University of Science and Technology, Taipa, Macao, China. [7]State Key Laboratory of Dampness Syndrome of Chinese Medicine, The Second Affiliated Hospital of Guangzhou Medical University, Guangzhou, China. [8]Guangdong Provincial Hospital of Chinese Medicine, Guangzhou, China. [9]Department of Neurology, The Second Affiliated Hospital of Guangzhou University of Chinese Medicine, Guangzhou, China. [10]Department of Microbiology and Immunology, Yong Loo Lin School of Medicine, National University of Singapore, Singapore, Singapore. [11]Immunology Programme, The Life Science Institute, National University of Singapore, Singapore, Singapore. [12]These authors contributed equally: Song Chen, Zongren Hu, Jianbang Tang. ✉e-mail: drguo@gzucm.edu.cn; miczy@nus.edu.sg; avenluo@gzucm.edu.cn

In our previous study, we showed that high temperature and humidity (HTH) can directly induce gut dysbiosis and minimal enteritis[6]. However, the underlying mechanisms of these effects are not fully understood. The aim of the present study was to bridge this knowledge gap by investigating the impact of HTH on mice in a controlled climate chamber. Our focus is understanding the interactions between bile acids (Bas) and the gut microbiota, abnormal lipid metabolism, and the suppression of plasma glucagon-like peptide-1 (GLP-1) levels in HTH conditions.

Notably, we explored the dysregulation of phosphatidylcholine (PthCho) metabolism and the induction of lysophosphatidylcholine (LPC) as potential explanations for previously observed gastrointestinal symptoms, such as diarrhea and enteritis. Moreover, GLP-1 plays a crucial role as a postprandial stimulus for pancreatic insulin secretion and a regulator of glucose homeostasis, making it a potential target for treating type 2 diabetes and obesity[7].

Building on our prior findings, this study further revealed that reduced lithocholic acid (LCA) levels can suppress GLP-1 through activation of the farnesoid X receptor (FXR). By elucidating these mechanisms, we aim to provide valuable insights into how climatic conditions, such as HTH, may influence metabolic diseases, thereby improving our understanding of the interplay among the environment, the gut microbiota, and human health.

## Results

### Plasma GLP-1 suppression in HTH

A schematic diagram of the experimental design is shown in Supplementary Fig. 1. We first observed a significant decrease in the plasma levels of GLP-1 and ghrelin in BALB/c mice exposed to high humidity (HH) or HTH for two weeks compared to those in mice in the normal control environment (NC group). This difference was more pronounced on day 14 (Fig. 1a, b). On the other hand, the insulin levels in the HH and HTH groups also decreased (Fig. 1c). The body weights, fasting glucose levels, peptide YY (PYY) levels, gastric inhibitory polypeptide (GIP) levels, and blood lipid levels (TG, HDL, LDL, and CHOL) of mice were comparable (Supplementary Fig. 2a–h). Blood glucose levels at all time points after glucose administration were significantly lower in the HTH group than in the control group (Supplementary Fig. 2i). This finding was further supported by lower area under the curve (AUC) values in the HTH group than in the control group (Supplementary Fig. 2j). As the suppression of GLP-1 in the HTH group was more pronounced than that in the HH group, we focused on HTH treatment for in-depth investigation of the mechanisms of GLP-1 suppression. We measured postprandial GLP-1 levels, and the results indicated that HTH conditions led to a reduction in postprandial GLP-1, with a significant decrease observed on day 14 (Fig. 1d). Additionally, HTH conditions led to a reduction in food intake, which became more pronounced on day 14 (Fig. 1e, f). Considering the critical role of GLP-1 in energy metabolism and inflammation, we investigated the underlying mechanisms of GLP-1 suppression in the HTH environment. High relative humidity reduces the ambient oxygen concentration in the air, and hypoxia inhibits GLP-1 production in vivo[8]. We wondered whether the increase in gut hypoxia caused GLP-1 suppression. However, we observed increased Hp-1 staining (a hypoxia probe indicating the degree of hypoxia) in the liver in the HTH group (Fig. 1g, i). Most importantly, unexpectedly, reduced Hp-1 staining in the ileum indicated a significant loss of hypoxia (Fig. 1h, j). This result

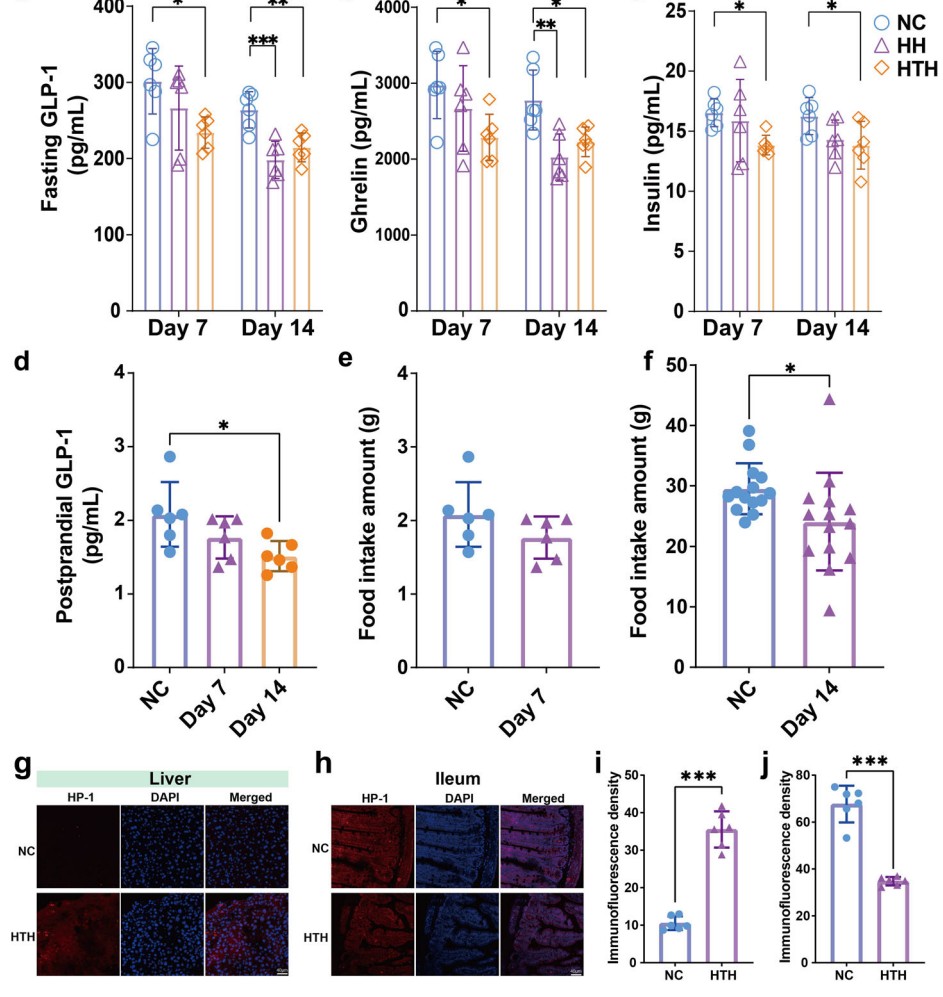

**Fig. 1 | The impact of environmental HH and HTH on bio-clinical parameters in mice. a** Plasma levels of fasting GLP-1 **b** ghrelin, **c** insulin, **d** postprandial GLP-1. $n = 6$ for each group. $*P < 0.05$, $**P < 0.01$, $***P < 0.001$ by one-way ANOVA. Error bars represent ±s.d. **e** Food intake amount of Day7. $n = 6$ for each group. Two-tailed Student's $t$ test. Error bars represent ±s.d. **f** Food intake amount of Day14. $n = 14$ for each group. $*P < 0.05$ by two-tailed Student's $t$ test. Error bars represent ±s.d. **g, i** The representative images of HP-1 staining of liver. **h, j** The representative images of HP-1 staining of ileum. Scale bars = 40 μm. $n = 6$ for each group. $***P < 0.001$ by two-tailed Student's $t$ test. Error bars represent ±s.d.

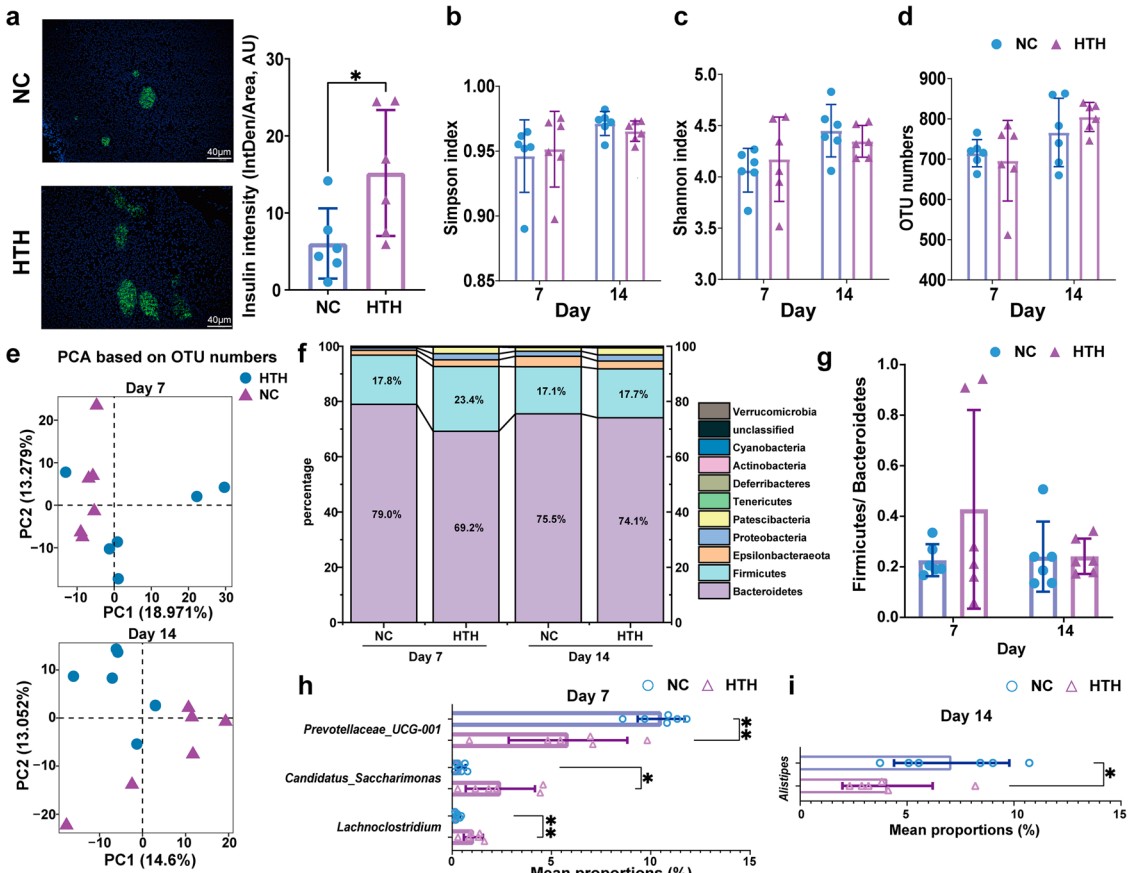

**Fig. 2 | Gut microbiota induced by environmental HTH. a** The pancreatic β cell mass, indicated by insulin immunofluorescence staining, was significantly expanded at 14 days after HTH treatment compared to NC group. The arbitrary units (AU) indicating the ratio of mean gray value to area were shown in columns. Scale bars = 40 μm. *$P < 0.05$ by two-tailed Student's $t$ test. Error bars represent ±s.d. Alpha diversity was applied to analyze the complexity of species of the NC group and HTH group. **b** Indices of Simpson and **c** Shannon, as well as **d** OUT numbers, are displayed. Error bars represent ±s.d. **e** PCA at the OTU level. The PCA of all the samples was based on the relative abundance of the OTUs. Percentiles represent the contributions of principal components to differences among samples. **f** The composition gut microbiota abundance at phylum level. **g** The ratio of *Firmicutes/Bacteroidetes*. **h, i** The gut microbiota with differential abundance at genera level at day 7 and day 14 after HTH treatment. $n = 6$ mice for each group. *$P < 0.05$, **$P < 0.01$ by two-tailed Student's $t$ test. Error bars represent ±s.d.

indicated that the liver became hypoxic, whereas the ileum lost its physiological hypoxia status.

## Alterations in the gut microbiota after HTH exposure

Although we have previously shown that HTH treatment induces gut dysbiosis, bacterial translocation, and enteritis[5], the rapid suppression of GLP-1 production after HTH exposure was unexpected. As immunofluorescence staining targeting insulin revealed an expanded area of β-cells in HTH-treated BABL/c mice (Fig. 2a), indicating augmented insulin synthesis, we deduced that the decrease in plasma insulin should be explained by deficient GLP-1-stimulated secretion. The intestinal foci, which were inhabited by L-cells, are profoundly impacted by HTH and play a pivotal role in HTH-induced abnormalities.

Therefore, we first investigated the impact of HTH on gut dysbiosis via 16S rRNA gene sequencing of fecal samples. We tested for overall differences in the microbial community structures between groups by measuring ecological parameters based on alpha diversity (Simpson and Shannon indices) and operational taxonomic unit (OTU) numbers. As shown in Fig. 2b–d, there was no significant difference in the mean values of either of the alpha-diversity indices or the number of OTUs between the HTH and NC groups. Principal component analysis (PCA) was conducted to investigate the temporal dynamics of the differences in OTU composition. As shown in Fig. 2e, the separation between the HTH and NC groups was more distinguishable on day 14 than on day 7, implying a progressive alteration in OTU composition.

Then, we compared the compositional distributions of the gut microbiota at the phylum level. As shown in Fig. 2f, g, the *Firmicutes/Bacteroidetes* ratio was greater in the HTH group than in the NC group on day 7. Moreover, the proportion of *Patescibacteria* phylum in the HTH group was greater on day 7 than that in the NC group (Fig. 2f). At the genus level, *Candidatus_Saccharimonas* and *Lachnoclostridium* abundance increased, while *Prevotellaceae_UCG-001* abundance decreased significantly on day 7 after HTH exposure (Fig. 2h) and *Alistipes* abundance decreased significantly on day 14 (Fig. 2i). Mendelian randomization (MR) analysis, which is a robust statistical method, utilizes genetic variants associated with an exposure as instrumental variables (IVs) to infer potential causal relationships between the exposure and an outcome from observed associations[9,10].

In this study, we utilized MR analysis to demonstrate that *Alistipes* may induce an increase in GLP-1, as evidenced by the data in Supplementary Table 1. We observed that in the HTH group, the levels of *Alistipes* were reduced, which corresponded with a decrease in GLP-1 levels. MR analysis suggested that the reduction in GLP-1 associated with HTH may be linked to alterations in the gut microbiome.

## Alterations in short-chain fatty acids (SCFAs) in HTH conditions

Although gut dysbiosis occurs rapidly after HTH exposure, its effect on GLP-1 production remains uncertain. We previously showed that HTH treatment caused the translocation of gut bacterial and elevated intestinal TNF-α levels[6], which has been related to impaired GLP-1 expression and

**Fig. 3 | Antibiotics and sodium butyric treatments on HTH-induced GLP-1 suppression. a** The effect of antibiotics cocktail treatment on the plasma level of GLP-1 and insulin. *n* = 6 mice for each group. ***P < 0.001 by two-tailed Student's *t* test. Error bars represent ±s.d. **b** The plasma levels of butyric acid at day 3 and day 10 after HTH treatment. *n* = 10 mice for each group. Error bars represent ±s.d. **c** The plasma levels of propionic acid at day 3 and day 10. **d** The effect of sodium butyrate administration of the GLP-1, **e** Insulin, and **f** fasting blood glucose level. **d, e** *n* = 10 mice for each group. **P < 0.01, **P < 0.001 by one-way ANOVA. Error bars represent ±s.d. **f** *n* = 9 mice in HTH group, *n* = 10 mice in NC, L.S.B and H.S.B group. *P < 0.05, **P < 0.01, **P < 0.001 by Kruskal–Wallis. Error bars represent ±s.d.

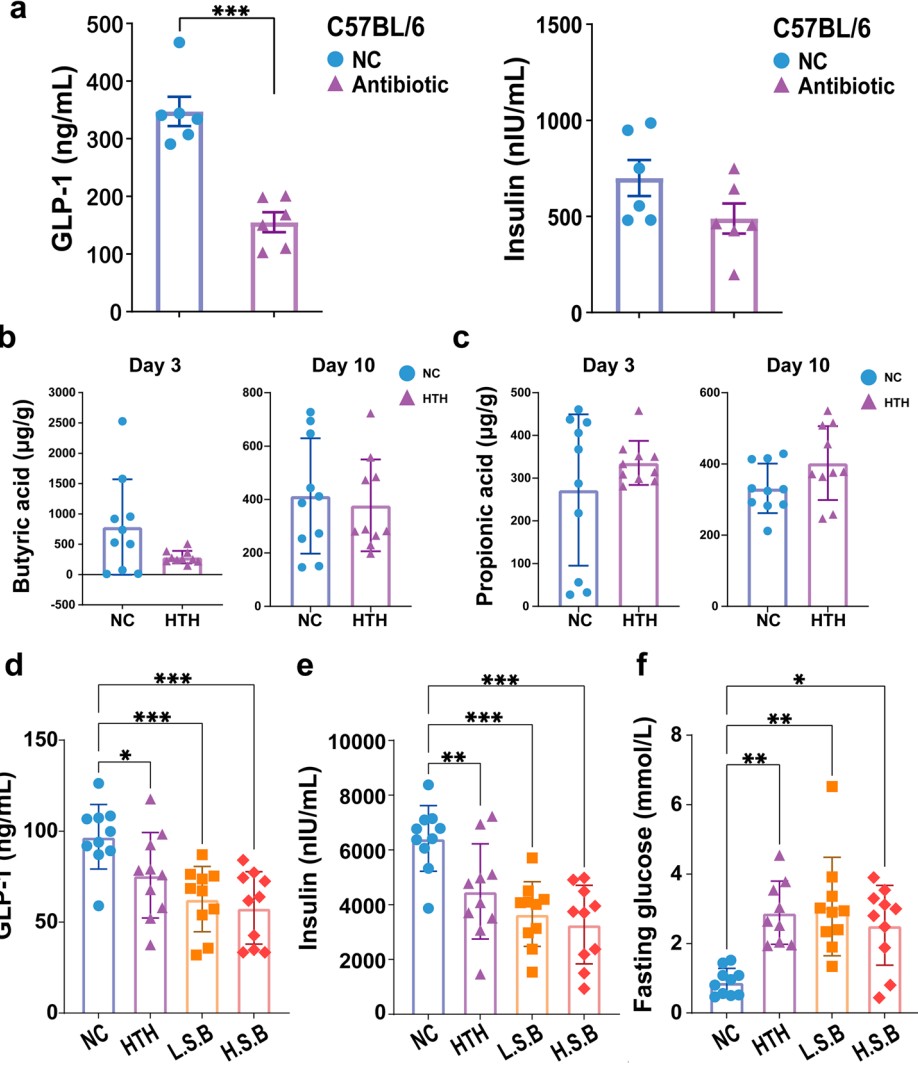

secretion[11]. In this study, we also found a significant increase in the expression levels of IL-6, MCP-1, and TNF-α after 2 weeks of exposure to HTH conditions (Supplementary Fig. 2k). Therefore, we wondered whether gut dysbiosis, as described above, could directly impact GLP-1 suppression through bacterial translocation. After oral administration of the antibiotic cocktail, we generated pseudo germ-free (PGF) mice (Supplementary Exp. 2, Supplementary Fig. 1). On day 14, the GLP-1 level of PGF mice was significantly decreased, accompanied by a slightly lower insulin level, compared to that of untreated mice (Fig. 3a).

We further performed a new experiment (Supplementary Exp. 3, as shown in Supplementary Fig. 1) and analyzed the collected samples to investigate the alterations after HTH exposure. The genera with differential abundance are shown in Supplementary Fig. 3. As shown in Fig. 3b, the butyric acid level on day 3 in the HTH group was lower than that in the NC group. However, the difference was temporary, and the reduction in butyric acid in the HTH group recovered on day 10. Moreover, the plasma level of propionic acid tended to increase on both day 3 and day 10, but the difference was not significant (Fig. 3c).

Then, we further evaluated the protective effect of butyric acid on HTH-induced GLP-1 suppression in vivo. Oral administration of both low-dose (L.S.B.) and high-dose (H.S.B.) sodium butyrate did not improve GLP-1 deficiency, decrease insulin levels, or increase FBG levels in HTH-treated mice (Supplementary Exp. 4, as shown in Supplementary Fig. 1) (Fig. 3d–f). Therefore, the decrease in butyric acid did not seem to explain the HTH-induced suppression of GLP-1.

## Alterations of BAs

We determined the hepatic expression of bile acid synthetic enzymes, including cholesterol 7α-hydroxylase (*Cyp7a1*), which is the rate-limiting enzyme in the classic pathway; oxysterol 7α-hydroxylase (*Cyp7b1*); sterol 12 alpha-hydroxylase (*Cyp8b1*), which is the key enzyme for the synthesis of cholic acid (CA); and sterol 27-hydroxylase (*Cyp27a1*), which is the initial enzyme in the alternative pathway. As shown in Fig. 4b–e, the expression of these enzymes decreased, and the expression of *Cyp7b1* decreased significantly, indicating significant suppression of the alternative pathway of BA synthesis; chenodeoxycholic acid (CDCA) synthesis was particularly affected.

To further elucidate the impact of BAs on the suppression of GLP-1, we conducted in vitro experiments to evaluate the effects of CDCA and its derivatives on GLP-1 production (Fig. 4a). Cultured Ncl-H716 cells, which are human L-cells, were treated with chenodeoxycholic acid (CDCA), taurochenodeoxycholic acid (TCDCA), and taurolithocholic acid (TLCA) at concentrations ranging from 25 to 100 μM. These treatments acted as agonizts and exhibited a dose-dependent effect on GLP-1 production. Notably, 100 μM LCA significantly increased GLP-1 production (Fig. 4f). These findings suggest that alterations in the composition of BAs may impair GLP-1 production. Then, we conducted a targeted metabolomics study using liquid chromatography−mass spectrometry (LC−MS) to assess the differences in the hepatic bile acid profile. The LCA level decreased in the HTH group compared to that in the NC group (Fig. 4g, Supplementary Data1). These findings suggest that the altered levels of LCA might explain

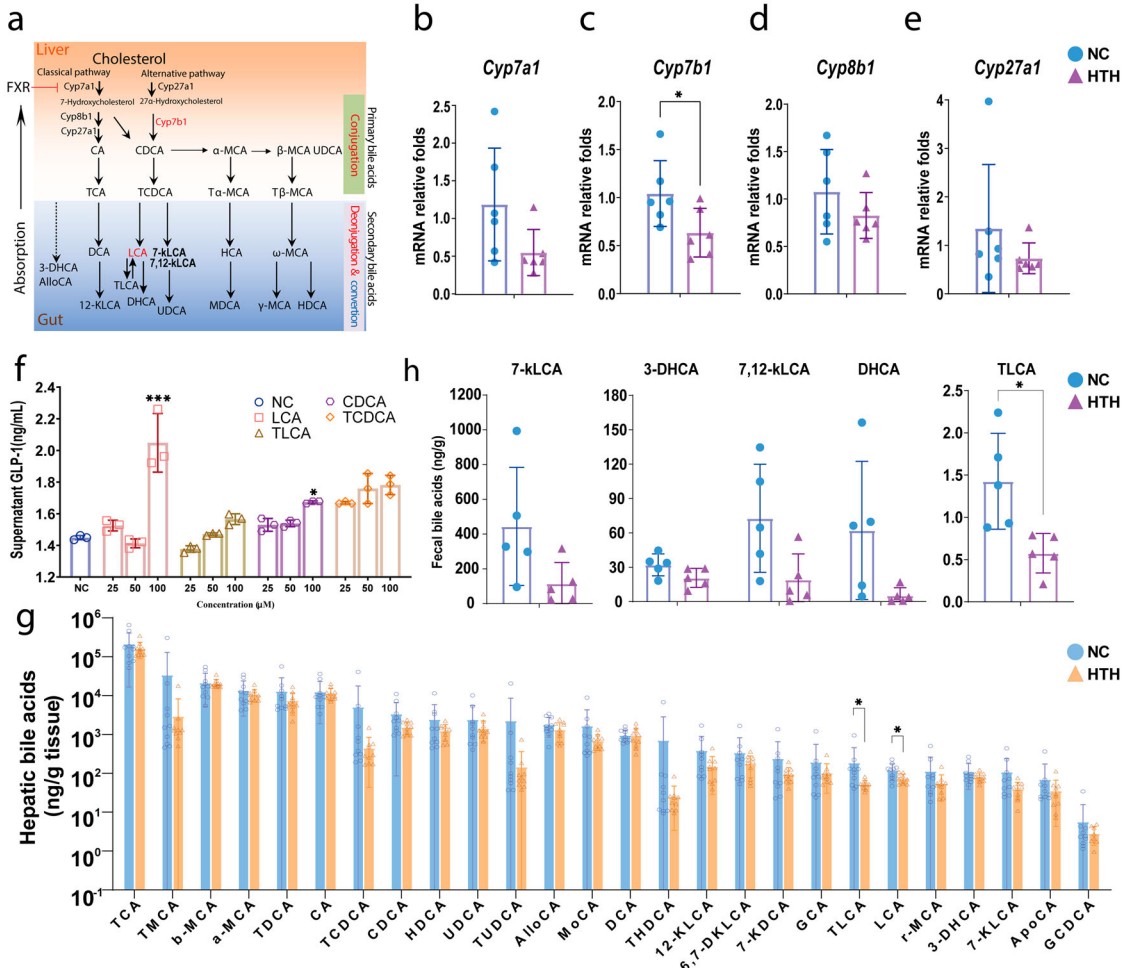

**Fig. 4 | BAs homeostasis in mice receiving persistent exposure of environmental HTH. a** The diagrammatic sketch of bile acids homeostasis and its regulatory effect on β-oxidation through FXR-ACC2 pathway. The hepatic expression of (**b**) *Cyp7a1*, (**c**) *Cyp7b1*, (**d**) *Cyp8b1*, and (**e**) *Cyp27* were shown in columns. **b–e** $n = 6$ for each group, *$P < 0.05$ by two-tailed Student's *t* test. Error bars represent ±s.d. **f** The level of GLP-1 after treated with LCA, TLCA, CDCA and TCDCA in vitro. $n = 3$ for each group, ***$P < 0.001$ by one-way ANOVA and Kruskal–Wallis test. Error bars represent ±s.d. **g** Targeted metabolomics analysis of BAs in liver. $n = 10$ for each group, *$P < 0.05$ by two-tailed Student's *t* test (LCA) and Mann–Whitney U test (TLCA). Error bars represent ±s.d. **h** Targeted metabolomics analysis of BAs in fecal. $n = 5$ for each group, *$P < 0.05$ by two-tailed Student's *t* test. Error bars represent ±s.d.

the suppression of GLP-1 induced by HTH, and we chose LCA for further research. We also evaluated the fecal BA profile. As shown in Fig. 4h, similar to that in the liver, we observed a significant decrease in TLCA and a marginal decrease in 7-kLCA, 7,12-kLCA, and DHCA in the HTH group.

**Ileal proteomic changes support intestinal FXR activation**

Considering the location of L cells and BA absorption, we selected ileal samples for proteomic analysis via tandem mass tagging (TMT) and liquid chromatography–mass spectrometry (LC-MS)/MS.

For comparison between the HTH and NC groups, proteins featuring a fold change of >1.2 or <0.83 and a *p* value of <0.05 were regarded as differentially expressed proteins (DEP). The hierarchical clustering heat map of the DEPs indicated that, large changes in protein levels were observed compared to the control group (Fig. 5a). The volcano plot in Fig. 5b shows the DEPs with the top three upregulated proteins and downregulated proteins were labeled. The top three enriched proteins identified in this study were perilipin 1, Slc10a2, and FABP6, whereas the top three deficient proteins were NCAPG2, MCPT2, and LCTL.

We noticed that the top enriched proteins, perilipin 1[12], SLC10A2, and FABP6[13], were encoded by genes directly positively regulated by FXR activation. These results implied a strongly activation of FXR pathway. Therefore, we compared the abundance of a protein panel encoded by FXR-

targeting genes. As shown in Fig. 5c, besides the above three proteins, others including BSAP, OST-α/β, MRP2, and ABCB4 were also significantly enriched. Therefore, the activation of ileal FXR signaling pathway should plays a pivotal role in the impact of HTH on bile acid metabolism.

In vitro experimental results revealed that, compared to the NC group, FXR inhibitors significantly increased the levels of GLP-1. Additionally, compared to the LCA group, the combination of LCA with FXR inhibitors led to a notable increase in GLP-1 levels. These in vitro findings suggest that LCA promotes GLP-1 production through the inhibition of FXR (Fig. 5d).

**Quantitative analysis of plasma metabolites**

The normalized intensities of whole metabolites detected by UHPLC-Q-TOF MS. With criteria of VIP (Variable Importance for the Projection)>1 and *P value* < 0.05, we identified 50 metabolites (21 downregulated and 29 upregulated) at day 3 (Supplementary Data 2), and 37 metabolites (25 upregulated and 12 downregulated) at day 10(Supplementary Data 3), with different abundance between NC and HTH groups under with Positive mode[14]. With PLS-DA plots, there was a clear separation between NC and HTH groups (Supplementary Fig.4). With a heatmap analysis, we showed that the samples in different group were clustered separately (Fig. 6a, b) implying a profound impact of HTH on the plasma metabolites.

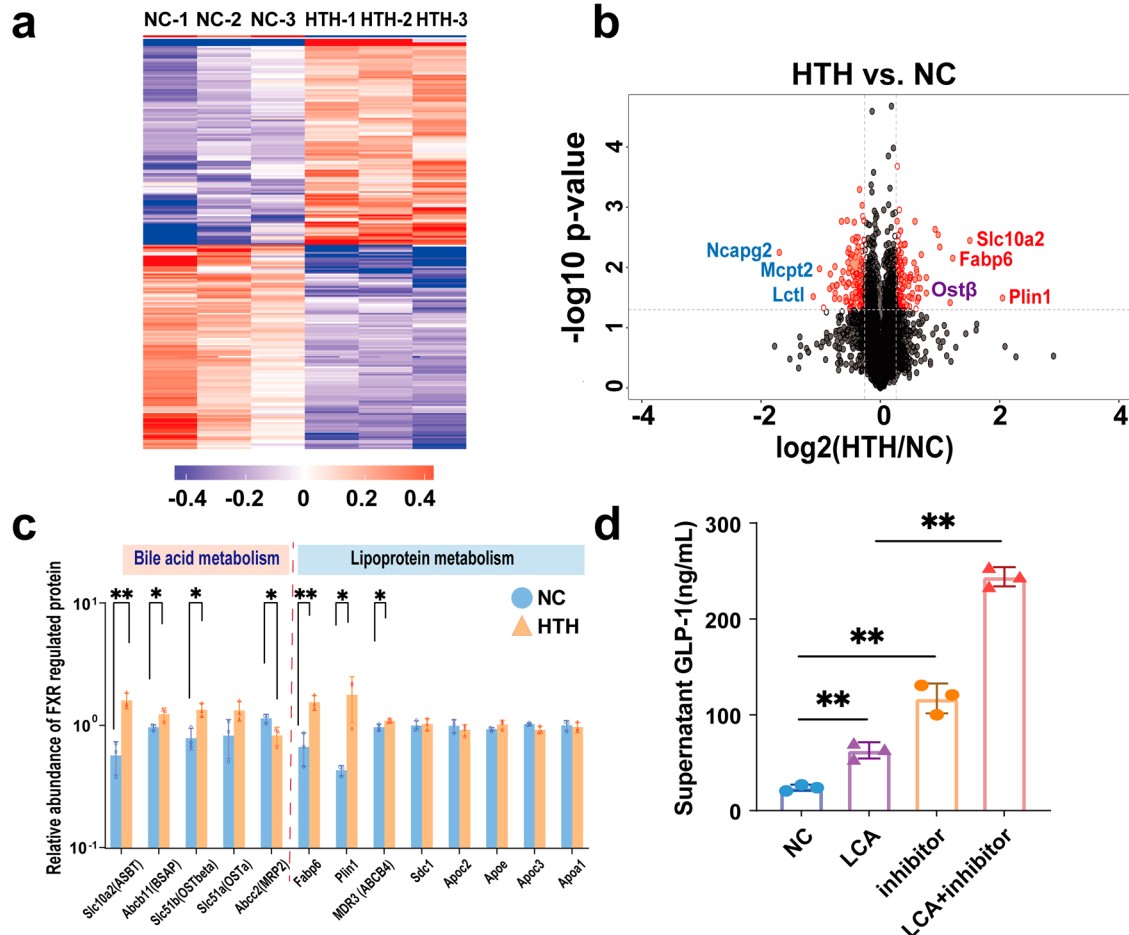

**Fig. 5 | Ileal proteomics analysis. a** Heat map indicates the differential expressed protein (DEP) with a criteria of change folds >1.2 or <0.83 and *p*-value < 0.05. **b** Volcano plot indicates DEPs with the above criteria and the top 3 enriched proteins and reduced proteins were labeled. **c** Relative abundance of FXR regulated protein. *n* = 3 for each group. *$P < 0.05$, **$P < 0.01$ by two-tailed Student's *t* test. Error bars

represent ±s.d. **d** The level of GLP-1 after treated with LCA, and FXR inhibitor in vitro. *n* = 3 for each group, **$P < 0.01$ by one-way ANOVA. Error bars represent ±s.d. Statistical significance was evaluated using *p*-values adjusted for multiple comparisons with the FDR method.

We observed a significant increase in the plasma levels of non-polar (glycine, alanine, valine, leucine, and methionine), aromatic (phenylalanine, tryptophan), and polar/uncharged amino acids (glutamine, threonine, asparagine, serine, tyrosine, and proline), as well as their derivates, at day 3 (Fig. 6c). Interestingly, at day 10, most of these amino acids in HTH group remained to be higher than that in NC group, except for amino acids predominantly present at the N-terminus of N alpha-acetylated proteins, like alanine, serine, and methionine. This specific alteration of amino acids profile implied an alteration on acetylation process.

The other notable alteration of metabolites profile is the abnormal phosphatidylcholine (PC) metabolism. As shown in Fig. 6d, the contents of serum PC in HTH group, including PC (16:0/16:0), SOPC (36:1), and DOPC (36:2), significantly decreased at day 3; whereas PC and SOPC rebound to higher than that in NC group at day 10. In consist with the alterations of PC, their metabolites, like long-chain fatty acids (palmitic acid and myristic acid), lysophosphatidylcholine (Lyso-PC,18:0, 18:1, 14:0), and PAF C18, as well as betaine, all decreased at day 3 and rebounded at day 10. Similarly, glycerol 3-phosphate, glycerol P-Cho, and phosphorylcholine were also decreased at day 3. This result implied an aberrant PC metabolism which emerged rapidly at day 3 and partially improved at day 10. Further, as shown in Fig. 6d, we observed an increase in 2-methylbutyroylcarnitine and carnitine accompanied with a decrease in palmitoylcarnitine at both day 3 and day 10. Therefore, we deduced that the carnitine shuttle was impaired. In consist with that, we showed a significant decrease in hepatic uncoupling protein 1 (Ucp1) expression, which is regulated by mitochondrial long-

chain fatty acids (Supplementary Fig. 5). However, hepatic expression of carnitine palmitoyl transferase 1α (Cpt1a) was upregulated (Supplementary Fig. 5), which is probably a feed-back activation responding to the deficient carnitine shuttle.

Moreover, the metabolomics analysis provided some other important clues for understanding the pathogenesis of HTH-related diseases. The serum taurocholate in HTH group is higher than that in NC group at day 3, and the hepatic tauro-conjugated BAs decreased simultaneously, indicated an augmented absorption of tauro-conjugated BAs at ileum. Further, we observed an increase in trimethylamine N-oxide (TMAO) at day 10, a metabolite transformed by gut microbiota like *Lachonoclostridium*, had been associated with the development of various diseases, such as cardio-vascular diseases[15].

**Correlation between metabolome and the intestinal microbiome**
Next, we tried to investigate whether specific microbial genera contributed to the alteration of these metabolites. With the criteria of LEfSe LDA > 2.0 and *P* < 0.05, we identified 23 genera with differential abundance from 16S rRNA gene sequencing, and 62 metabolites with differential abundance based on the criteria of VIP > 1 and *P* < 0.05, as well as 27 kinds of BAs. With *Spearman* correlation coefficient analysis, we identified a significant correlation between these metabolites, BAs and the specific gut microbial genera. The heatmap presented the correlation coefficient values in colors with various degree of red (r > 0, indicating a positive correlation) or blue (r < 0, indicating a negative correlation), respectively. As for the BA profile,

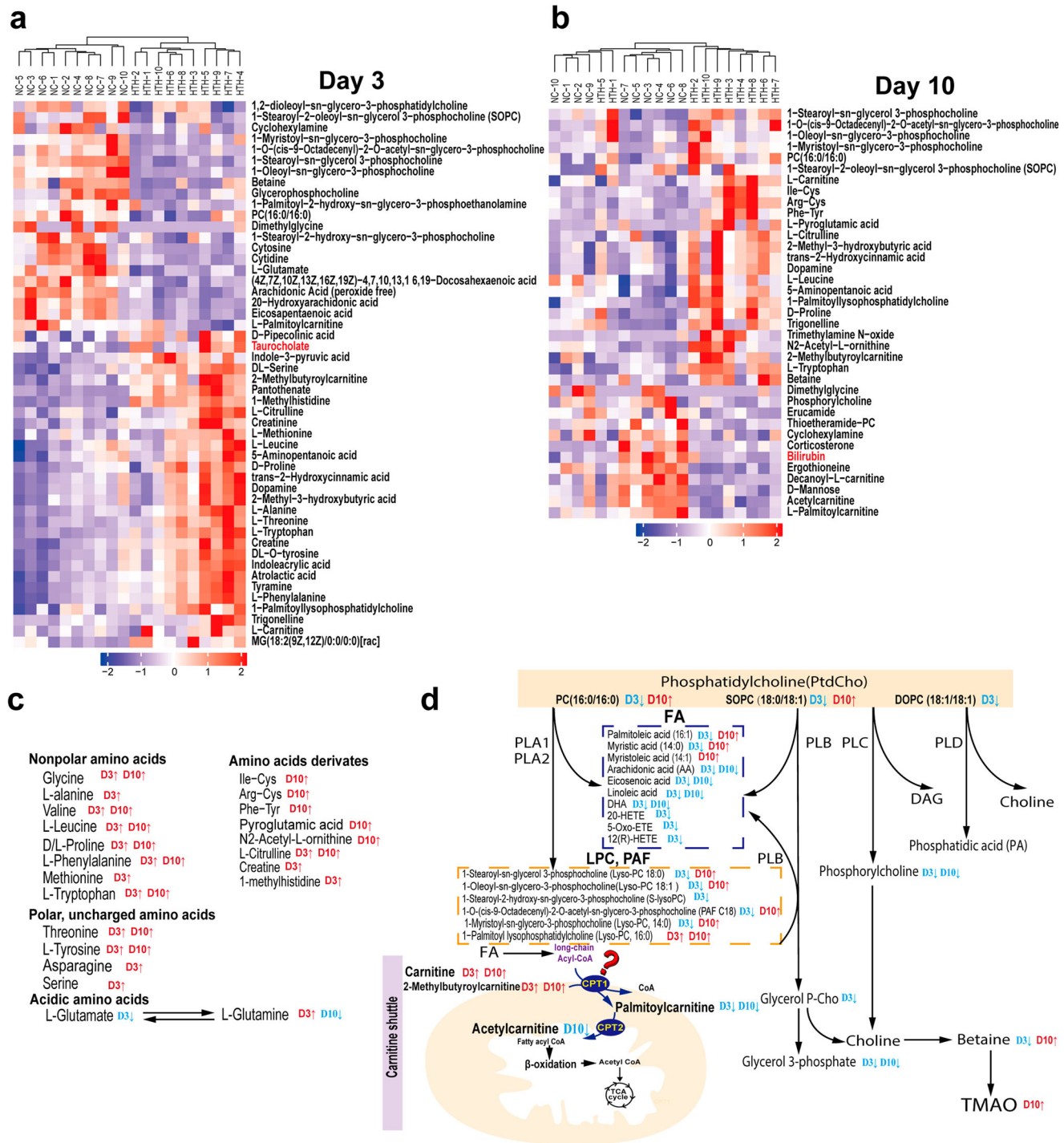

**Fig. 6 | Metabolomic Analysis of plasma samples. a**, **b** Heatmap of metabolites with differential abundance between HTH and NC groups. **c** Amino acids. **d** Schematic diagram of the metabolic pathway to obesity induced by a high-fat diet based on serum and liver metabolites analysis. The metabolites analyzed by LC-MS and GC-MS are shown in color; red represents increased metabolites, green represents decreased metabolites, yellow represents no change, and the open circles represent no detected metabolites. Statistical significance was evaluated using *p*-values adjusted for multiple comparisons with the FDR method.

we focused on the contributors affecting LCA/TLCA and their derivates, like 7k-LCA and 12-kLCA. As shown in Fig. 7a, these BAs were positively related with *Parabacteroides*, *Bacteroides*, especially *Parabacteroides*, and negatively corelated with *Ruminococcaceae* family (*Ruminococcaceae_UCG_005*, *Ruminococcaceae_NK4A214_group*, *Ruminococcaceae_UCG_010*) and *Butyricimonas*.

As for the relationship between gut microbiota and metabolites identified from non-target metabolomic analysis, we focused on the PtdCho metabolism involving 6 metabolites including PC(16:0/16:0), Oleoyl-sn-glycero-3-phosphocholine, 1-Myristoyl-sn-glycero-3-phosphocholine, 1-Palmitoyllysophosphatidylcholine, 1-Stearoyl-sn-glycerol-3-phosphocholine, 1-O-(cis-9-Octadecenyl)-2-O-acetyl-sn-glycero-3-phosphocholine) as the Lyso-PC and PAF could mediated inflammation and contribute to the pathogenesis of a wide range of diseases. As shown in Fig. 7B, among these 6 metabolites, *Ruminococcaceae_NK4A214_group*, *Butyricimonas*, and *Tolumonas* were the most relevant genera which correlated were positively related with 4, 3, and 2 of 6 metabolites with a threshold of *P* < 0.01 (difference with 0.01 < *P* < 0.05 was ignored for clarity). Moreover, other five

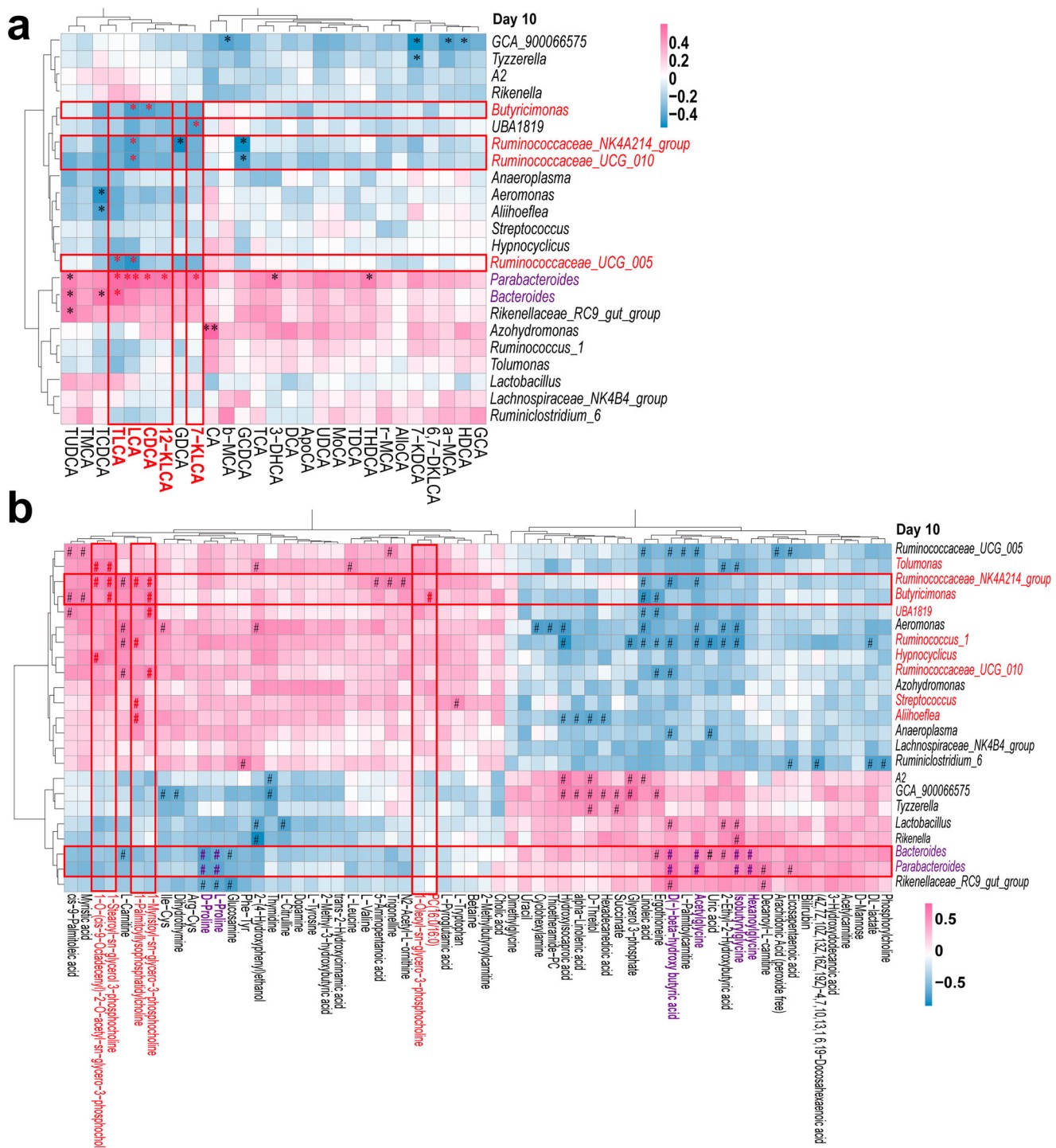

**Fig. 7 | The Spearman relationship between different bacteria genera and plasma metabolites. a** Heatmap indicating the relationship between different triggering bacteria and plasma metabolites. The coefficients of correlation (r) were represented with various degree of red (indicating a positive correlation with r > 0) or blue (indicating a negative correlation with r < 0), respectively. *P < 0.05; **P < 0.01.

**b** Heatmap indicating the relationship between different triggering bacteria and plasma bile acids. #P < 0.01 (Note: significance with 0.01 < P < 0.05 was ignored for clarity. Statistical significance was evaluated using p-values adjusted for multiple comparisons with the FDR method.

genera were also related to the levels of lyso-PC or PAF (Fig. 7b). Interesting, *Parabacteroides* and *Bacteroides*, positively correlated with LCA/TLCA, were negatively related with metabolites clustered in proline, Hexanoylglycine, Isobutyrylglycine, Acetylglycine, and D (-)-beta-hydroxy butyric acid (in purple). These five metabolites could probably be involved in the same metabolic pathway.

## Discussion

Environmental HTH is associated with discomfort and several diseases. In this study, we confirmed that HTH rapidly induced a remarkable disturbance in lipid metabolism, bile acids-gut microbiota interaction, and GLP-1 production. Enteroendocrine L-cells respond to various secretagogues through a range of sensing and signaling pathways. The antibiotics

cocktail treatment aggravated HTH-induced GLP-1 suppression implying a protective role of gut microbiota in GLP-1 production. Gut microbiota-derived SCFAs act as signaling molecules stimulating GLP-1 secretion by binding to their transmembrane receptors FFAR2 (GPR43) and FFAR3 (GPR41) on colonic L-cell response[16]. Moreover, SCFAs modulate excit-ability and proliferation of enteric neurons[17,18] which play critical role in GLP-1 regulation. In this study, HTH caused a distinct alteration of gut microbiota charactered as increased *Firmicutes/ Bacteroidetes* ratio, decreased *Prevotellaceae_UCG-001* accompanied with increased *Lachno-clostridium* and *Candidatus_Saccharimonas* at day 7. *Prevotellaceae_UCG-001*, involved in saccharolytic fermentation, probably contributed to the decreased plasma butyric acid level in our study. However, the decreased butyric acid recovered rapidly and could not explain the GLP-1 suppression as butyrate supplement could not alleviate GLP-1 decline.

The gut microbiota plays a significant role in the synthesis of bile acids[19]. We speculated that the alteration of BAs transformed by gut microbiota[20,21], should be responsible for the decreased GLP-1 in HTH group. In this study, the rate-limiting enzymes of alternative BAs synthesis pathway was inhibited. In vitro experiments, we showed that LCA increased GLP-1 secretion. In vivo studies revealed that HTH conditions can lead to a reduction in both LCA and GLP-1 levels. Further, we tried to investigate the intrinsic mechanisms of GLP-1 suppression, with ileum proteomics analysis.

GLP-1 is produced by L-cells in response to nutrients, microbial fac-tors, bile acids and short-chain fatty acids, as well as stimuli from enteric cholinergic and adrenergic receptors[7]. Among these factors, bile acids and gut microbiota are reciprocally regulated and collaboratively control GLP-1 production mainly through the binding of BAs to their receptors, FXR[22]. The activation of FXR decreases GLP-1 secretion[23]. In this study, we showed that the ambient HTH affected the crosstalk between gut microbiota and BAs and thereby caused an aberrant FXR activation and GLP-1 suppression. With proteomics technique, we showed that environmental HTH induced intestinal FXR activation, dominated the BAs mediated GLP-1 regulation. Perilipin 1, FABP6 and Ost-β, direct target genes of FXR, were enlisted as the top-10 upregulated proteins. In the intestine, FXR controls the absorption of bile acids, lipids, vitamins, certain drugs, and other xenobiotics, besides regulating GLP-1 production. The remarkable increase in the protein contents of SLC10A2, FABP6, and OSTβ indicating an augment reab-sorption of bile acids[24]. In addition, as bile acids synthesis in the liver is negatively regulated by FXR-dependent mechanisms[25], which explained the reduced hepatic BAs pool. FXR activation inhibits intestinal cholesterol absorption with the FXR agonist GSK2324 reduced absorption and selective decreases in fatty acid synthesis[24,26].

Although BAs (like CA, CDCA, and CA) are FXR agonizts[27], and all of them reduced slightly after HTH treatment, the intestinal FXR signaling pathway was activated. The reasonable explanation for which is the sig-nificantly decrease of LCA. It has been described that LCA binds to FXR with a higher affinity than CDCA in the FXR co-activator association assay and acts as a pure antagonist with no detectable agonist activity[28]. We also showed that LCA in the physiological concentration acts as an antagonist for GLP-1 production. Therefore, we confirmed that the alteration of BAs composition, especially the reduced LCA, contributed to HTH induced GLP-1 suppression through FXR activation.

Interestingly, in addition to maintaining bile acids, cholesterol and glucose homeostasis, FXR also regulates fatty acid β-oxidation (FAO). FXR agonist decreases lipid accumulation by promoting hepatic fatty acid oxi-dation in *db/db* mice[29]. Activation of FXR significantly increased fatty acyl-CoA hydrolysis (Acot1) and decreased FAO-associated mRNAs, resulting in reduced levels of total acylcarnitines and relative accumulation of long/medium chain acylcarnitines and fatty acids in liver[30]. Therefore, the FXR activation could HTH induced arrest carnitine shuttle and FAO.

Further, metabolomic analysis provided us more clues for under-standing the pathogenesis of HTH caused abnormalities. First, we observed a blockage of carnitine shuttle, a biological process facilitates the transport of long-chain fatty acids from the cytosol into the mitochondrial matrix for

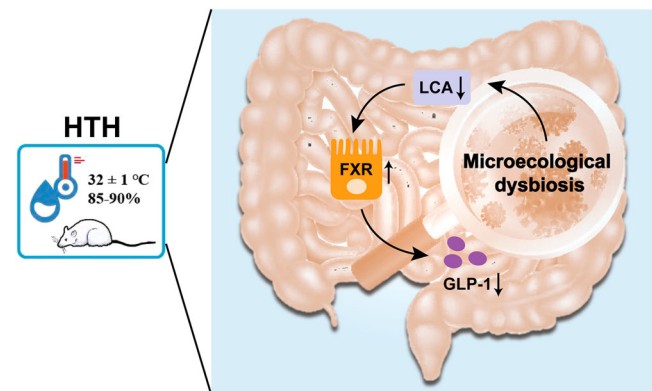

**Fig. 8 |** The diagram of the results for this study.

subsequent β-oxidation. This abnormality might probably inhibit fatty acids oxidation. Carnitine also binds acyl residues deriving from the intermediary metabolism of amino acids and help in their elimination functioning as a scavenger. Second, we showed a decrease in PtdCho and their derivates rapidly after HTH exposure. However, partial PtdCho, Lyso-PC and PAF in HTH group rebounded to be higher than control at 10 days. As the lyso-pc and PAF, probably gut microbiota-mediated lysophosphatidylcholine, can augment inflammation[31,32], promotes colitis, as well as other inflammatory diseases. In consist, we previously described that HTH caused enteritis. Moreover, we observed elevated plasma TMAO, which are transformed from choline by specific bacterial genera, and associated with high athero-sclerosis burden[31–33]. Therefore, these aberrant levels of metabolites could explain the high prevalence of many metabolic diseases in HTH environment.

We further performed a correlation analysis to reveal the relationship between BAs, metabolites, and gut microbiota. We first identified bacteria *Parabacteroides* and *Ruminococcaceae* family which is negatively and positively correlated with LCA levels, respectively. *Parabacteroides* dis-tasonis alleviates obesity and metabolic dysfunctions *via* production of succinate and secondary bile acids[34]. Administration of *Parabacteroides* distasonis leads to increased bile salt hydrolase (BSH) activity, inhibition of intestinal FXR signaling and decreased TCDCA levels in liver[35]. *Rumino-coccaceae* families perform the subsequent 7α-dehydroxylation of CA and CDCA to respectively generate DCA and LCA, has been associated with intermittent hypoxia and hypercapnia conditions[36]. *Lachnoclostridium* is a genus of gram-positive, obligate anaerobic, spore-forming, motile bacteria. Organisms in this genus can grow in moderate 'mesophilic' as well as in extremely high 'thermophilic' temperatures[37], which could confer it a fitness advantage in HTH. *Lachnoclostridium* is described with higher microbial capacity for bile acid conversion through a 7α-dehydratase process[38], explained the robust increase in LCA abundance. Moreover, *Lachnoclos-tridium* produces trimethylamine (TMA), which was turned into trimethylamine-N-oxide (TMAO) in liver, and mediate diseased including cardiovascular disease, heart failure, kidney diseases, metabolic syndrome[15,39].

In summary, we demonstrated that exposure to HTH disrupts the gut microbiome, leading to a reduction in LCA levels. This decrease in LCA triggers an upregulation of FXR expression, ultimately resulting in dimin-ished GLP-1 secretion (shown in the Fig. 8). However, our research does not explicitly unravel the mechanistic pathways which govern the changes of the gut microbiome and BAs upon HTH. In future research, we aim to conduct more focused experiments to elucidate these direct mechanisms.

## Methods
### Ethical approval
We have complied with all relevant ethical regulations for animal use. Male BALB/c and C57BL/6 mice, aged 6 weeks, were used in the research. All experiments were approved by the Animal Experimental Ethical Committee

of the Traditional Chinese Medicine Hospital of Zhongshan (no. 2018007). All related procedures were performed in accordance with the recommendations of the National Institutes of Health Guide for the Care and Use of Laboratory Animals [National Research Council, Guide for the Care and Use of Laboratory Animals. (2011)].

## Experiment of high temperature and high humidity treatments

The schematic diagram of experiment design was shown in Supplementary Fig.1. Male BALB/c and C57BL/6 mice (six weeks of age) were obtained from and maintained under specific pathogen-free (SPF) conditions at the Laboratory Animal Research Center of Traditional Chinese Medicine Hospital (Zhongshan City, Guangdong Province, China). All study animals were housed in sterile cages in a room maintained at $20 \pm 1\,°C$, with an average humidity of $60 \pm 2\%$ and a 12/12-h light: dark cycle. The mice were given access to water and chow ad libitum. Following two weeks of acclimation, mice were randomly assigned to groups (Supplementary Fig.1). Mice in the NC group were maintained as described above, whereas mice in the HTH and HH groups were regularly transferred into sterilized climate chambers (model: RXZ-158A, Ningbo Jiangnan Instrument Factory, China) located in the same room. The climate chambers were set to temperatures of $32 \pm 1\,°C$ and relative humidity (RH) levels of 85–90% (HTH group) or solely 85–90% RH (HH group) over 14 consecutive days as previous described[6,40]. Food and drinking water were sterilized and renewed daily to avoid contamination. In our study, mice from different groups were housed in separate cages to avoid cross-contamination. Specifically, in Supplementary Exp 1, each group consisted of 6 mice, and all these mice were housed together in the same cage. Similarly, in Supplementary Exp 3, each group had 5 mice, and these mice were also housed together in a single cage.

## Antibiotic cocktail-mediated perturbations of the gut microbiome

To determine the role of commensal microflora in GLP-1 suppression, mice were depleted of all detectable commensals by an oral administration of antibiotic cocktail, including ampicillin (1 mg/mL), metronidazole (0.25 mg/mL), neomycin trisulfate (1 mg/mL), and vancomycin (0.5 mg/mL). Antibiotics were resolved in autoclaved drinking water. This "pseudo germ-free mice" model has been widely used to assess the impact of alterations in the microbiome in the fields of immunology and metabolism[30]. Water consumption was monitored every day.

## Oral glucose tolerance test

OGTT was performed on mice fasted overnight for 12 h. The mice were orally fed glucose (2 g/kg as a 20% glucose stock solution). Blood was drawn from the tail vein and the glucose levels were measured using an ACCU-CHEK Advantage llGlucose Monitor (Roche, IN, United States) at 0 (before glucose load), 15, 30, 60, and 120 min after glucose administration. Data were expressed as the absolute values of blood glucose concentrations.

## Food intake

Daily food intake was measured by harvesting the food from the grill of the cage each day over a period of 7 and 14 days, thereby calculating the daily food intake.

## Cell culture

NCI-H716 cells (ATCC, CCL-251) was maintained in RPMI-1640 (Gibco, Life Technologies) supplemented with 10% fetal bovine serum, 2 mM L-Glutamine, 50 IU/mL penicillin and 50 μg/mL streptomycin in a humidified incubator at $37\,°C$, 5% of CO2. For GLP-1 secretion studies, $4 \times 10^5$ NCI-H716 cells were incubated 24 h with appropriate concentration of CDCA, TCDCA, LCA, TLCA (all purchased from Aladdin Inc., China, with purity >99%) and GUDCA (Sigma-Aldrich) or vehicle. Supernatant was centrifuged and stored at $-80\,°C$ until used.

The NCI-H716 cell line is not listed in the International Cell Line Authentication Committee (ICLAC) database of commonly misidentified cell lines (http://iclac.org/databases/cross-contaminations). The cell line was

obtained from American Type Culture Collection (ATCC, USA) and the cell line was authenticated using STR profiling, which confirmed its identity and purity.

## Quantitative reverse-transcription PCR

Total RNA was extracted using Trizol (Invitrogen, 15596018CN, USA) and a color reverse transcription kit (with gDNA Remover) (EZBioscience, A0010CGQ, China) following the manufacturer's instructions. qPCR was conducted using 2× Color SYBR Green qPCR Master Mix (ROX2 plus) (EZBioscience, A0012-R2, China). The qPCR protocol involved an initial step at $95\,°C$ for 5 min, followed by 40 PCR cycles ($95\,°C$ for 10 s and $60\,°C$ for 30 s), and finally a step at $95\,°C$ for 15 s and $60\,°C$ for 1 min to obtain the dissolution curve and terminate the reaction. The primers used in this experiment are listed as below: *Cyp7a1*:forward: CTGCTACCGAGTG ATGTTTGAA, reverse:GAAAGTCGCTGGAATGGTGTT; *Cyp7b1*:forward: GCCTCTCTAGCAAACACCATTCCAG, reverse:GAAAGTCGC TGGAATGGTGTT; *Cyp8b1*:forward:ATACCCTGAAGATGTCCAGTG G,reverse: AATTTCGTCACGCAGGGCTTCC; *Cyp27a1*:forward: GTGC CCCGCTCTTGGAGCAA, reverse:CCTTCCGTGGTGAACGGCCC;

## Treatment with sodium butyrate in mice

Sodium butyrate (99% purity, purchased form Aladdin Inc, China) was dissolved in water and intragastrically administrated to mice in doses of 400 mg/kg (high dose, H.S.B) or 80 mg/kg (low dose, L.S.B) for 14 consecutive days.

## Blood biochemical analysis and enzyme-linked immune-sorbent assay (ELISA)

At predetermined time points, the mice were fasted for approximately 16 h and then euthanized via cervical dislocation. Next, EDTA-treated whole blood samples were collected. Plasma was separated and immediately subjected to biochemical analysis using a Hitachi 7080 biochemical analyzer (Japan). Aliquots of plasma were stored at $-80\,°C$ until use. Mouse insulin, glucagon, GLP-1, and ghrelin, ELISA kits were purchased from AndyGene Biotechnology Co. LTD (Beijing, China). These parameters were used to quantify the respective levels in the plasma in accordance with the manufacturer's instructions. A Varioskan LUX microplate reader (Thermo Fisher, USA) was used for the assay, and a 4-parameter logistic regression was used for data analysis. Two replicates were performed for each sample. The results were then averaged.

## 16S rRNA gene sequence analysis

Fecal samples were harvested from each mouse through abdominal pressing and collected in sterilized tubes. Samples were immediately frozen and maintained at $-80\,°C$ until processing for bacterial DNA isolation and extraction, which was performed in accordance with previously reported methods[41]. Briefly, the bead-beating method was used to collect DNA, which was dissolved in a Tris-EDTA (TE) buffer after extraction using phenol and chloroform.

Six fecal samples per group was performed for 16S rRNA gene sequence. Gene amplification, cloning, and sequencing of the bacterial 16S rRNA gene PCR products were conducted in a laboratory maintained by BGI-ShenZhen (Beijing Genomic Institute, Shenzhen Huada Gene Institute, China). PCR amplification of the V5-V4 regions of the bacterial 16S rRNA gene[42,43] was conducted using universal primers (515F 5′-GTG CCAGCMGCCGCGGTAA-3′ and 806R:GGACTACHVGGGTWTCTA AT;), which incorporate unique sample barcode sequences. To generate an amplicon library, 20 ng of each genomic DNA sample, 1.25 U of Taq DNA polymerase, 5 μl of 10× Ex Taq buffer, 10 mM dNTPs (all reagents from TaKaRa Technology Co., Ltd, Dalian, China), and 40 pmol of primer mix were added to a 50 μl reaction mixture. The PCR conditions were as follows: a 5 min initial denaturation at $95\,°C$, 28 cycles of denaturation at $95\,°C$ (30 s), annealing at $55\,°C$ (30 s), and elongation at $72\,°C$ (45 s), followed by a final extension at $72\,°C$ for seven min. The PCR products were purified using magnetic beads (Axygen Biosciences, Union City, CA, USA), and the

concentration of the amplicon library was estimated using a 2100 Bioanalyzer System (Agilent Technologies Inc., Waldbronn, Germany). Equal amounts of amplicons from each sample were pooled. Sequencing data were retrieved from the European Bioinformatics Institute (accession number: ERP120131).

Raw data were treated using an in-house pipeline developed based on Mothur v.1.31.2[44]. The primers were removed, low-quality sequences (average quality scores of the 30 bp window <20) were truncated, and all high-quality reads from individual samples (lengths >250 bp) were pooled. Thereafter, operational taxonomic units (OTUs) were clustered with a 97% identity cutoff using USEARCH (v7.0.1090)[45]. The relative abundances of the OTUs were then calculated and those lower than 0.001% were removed. This was followed by an analysis of the I distribution in each sample. Both weighted and unweighted UniFrac distance analyses were performed based IOTU abundances and the phylogenetic tree, and a principal component analysis (PCA) was conducted based oIhe OTU abundance profiles obtained using customized R (3.0.2) scripts. Base on the OTU abundances and taxonomic annotations, we obtained relative abundance profiles at the phylum, class, order, family, genus, and species levels[46]. Alpha diversity, beta diversity, and rarefaction curve analyses were conducted based on tIrelative OTU abundance table. The Kruskal-Walli's test was conducted to explore the enrichment of bacterial species and functions in different subgroups. The FDR was used to control type I errors and enable the identification of the most reliable candidates for multiple group comparisons.

## Measurement of SCFAs

SCFA concentrations in the plasma were measured using high performance liquid chromatography (HPLC). A mixture of 100 µl plasma and 200 µl of crotonic acid (0.5 mM), an internal standard, were pre-labeled with 2-nitrophenylhydrazide using a Short- and Long-Chain Fatty Acid Analysis Kit (YMC Co., Ltd., Kyoto, Japan). The SCFA derivatives were extracted with n-hexane and diethyl ether, and subsequently evaporated to dryness. The residue was reconstituted with methanol, and filtered through a 0.2 µm syringe filter. 10 µL of this filtrate was injected into an HPLC system with a YMC-Pack FA column (250 × 6.0 mm; YMC Co., Ltd.).

The HPLC system (JASCO, Tokyo, Japan) consisted of two pumps (PU-980), a column oven (CO-965), an autosampler (AS-950), a UV-VIS detector (UV-970) and an integrator (LCSS-905). We performed HPLC under the following conditions. The column oven temperature was 50 °C, the mobile phase consisted of acetonitrile-methanol-water (30:16:54 v/v, pH 4–5 adjusted by 0.01 N HCl), the flow rate was 1.2 ml/min and the eluate absorbance was monitored online at a wavelength of 230 nm. To construct SCFA calibration curves, we prepared standards of acetic, propionic, butyric, isobutyric, valeric, isovaleric and hexanoic acids at 0.1–5.0 mM (0.1, 0.2, 0.5, 1.0, 2.0 and 5.0 mM). The correlation coefficients of the calibration curves were 0.9954–0.9998. Recovery tests were performed by adding known amounts (10 µmol) of each SCFA to 10 ml aliquots of calibration standard solution (1.0 mM); the recovery ranged from 95.9 to 118.0%. All procedures of the SCFA concentration analysis were performed in duplicate.

## Bile acid profiling by liquid chromatography-tandem mass spectrometry (LC-MS/MS)

Liver samples (30 mg) were homogenized with 200 µl pre-cooled ultrapure water, 800 µL pre-cooled methanol, and 10 µL internal standard, then mixed thoroughly by vortex, followed by incubation at −20 °C for 20 min to allow the protein precipitation. Then, the samples were centrifuged at 14,000 rcf for 15 min at 4 °C, and the supernatant was collected and dried under vacuum. Methanol-water (100 µL, 1:1, v/v) was added for reconstitution. The standard was diluted to a series of gradient concentration standard working solutions with methanol aqueous solution, the standard curve solution was prepared according to the above method, and a standard curve was established using the isotope internal standard method. The samples were analyzed using an Acquity UPLC system (Waters Ltd.) coupled online to a 5500 QTRAP mass spectrometer (AB SCIEX, USA). The samples (2 µl) were injected onto an ACQUITY UPLC BEH C18 1.7 µm, 2.1 mm × 100 mm)

column (Waters Ltd.). The samples were eluted at a flow rate of 250 µL/min with phases A (0.1% formic acid in water) and B (methanol). The separation was performed as follows: linear gradient from 60 to 85% B (0–15 min), isocratic at 85% B (15–17 min), linear gradient from 85 to 60% B (17–17.1 min), and isocratic at 60% B (17.1–20 min). The column temperature was 45 °C. A QC sample was used for each set of experimental samples in the sample queue to test and evaluate the stability and repeatability of the system. The samples were separated using a Waters UPLC system. Mobile phase: 0.1% FA aqueous solution was used in phase A, with methanol in phase B. The sample was placed in an 8 °C autosampler with a column temperature of 45 °C, a flow rate of 250 µL/min, and an injection volume of 2 µL. The relevant liquid phase gradient is as follows: 0–7 min, phase B linearly changed from 60 to 70%; 7–15 min, and phase B linearly changed from 70 to 85%. At 15–17 min, phase B was maintained at 85%, then from 17–17.1 min, the phase changed linearly from 85 to 60%. From 17.1–20 min, phase B was maintained at 60%. A QC sample was used for each set of experimental samples in the sample queue to test and evaluate the stability and repeatability of the system. Mass spectrometry was performed as follows: source temperature: 550 °C, ion source Gas1 (Gas1): 55, Ion Source Gas2 (Gas2): 55, curtain gas (CUR): 40, and ionSapary Voltage Floating (ISVF): -4500 V. The MRM mode was used to detect the transitions to be measured.

The raw LC-MS/MS data were analyzed using Multiquant software to obtain the calibration equations and quantitative concentrations of each BA in the samples. Differences in BA measurements between the groups were analyzed using Student's $t$ test, with $p < 0.05$ considered significant. The regression equation is as follows: derived from the standard curve, the relative deviation of the sample's lower limit of quantification (LLOQ), was ≤20%, the relative deviation of the other concentrations and quality control (QC) relative standard deviation (RSD) was ≤30%, and the square of the correlation coefficient R was >0.99. The contents of each test sample are as follows: calculated from the standard curve. Spearman's correlation analysis was used to analyze the correlation between fecal microflora and metabolomics. Using R software (version 3.3.1) to generate graphics, we obtained results with $p < 0.05$, representing statistical significance.

## TMT-based quantitative proteomic analysis

Proteins were extracted from minced ileal samples with an ice-cold SDT (4% SDS, 100 mM Tris/HCl pH7.6, 0.1 M DTT) lysis buffer60. Total protein concentrations were determined using a bicinchoninic acid (BCA) protein assay kit (Pierce, Rockford, IL, USA). An appropriate amount of protein from each sample was digested with trypsin using the filter-aided proteome preparation (FASP) method. Peptide concentrations were measured at OD280. Following trypsin digestion, the peptides were desalted using a Strata X C18 SPE column (Phenomenex), and 100 µg of each peptide was processed according to the manufacturer's protocol of the TMT labeling kit (Thermo Fisher Scientific, Torrance, CA, USA).

The pooled samples were then fractionated via high-pH reverse-phase HPLC using an Agilent 300Extend C18 column (5 µm particle size, 4.6 mm ID, 250 mm length). Briefly, peptides were first separated with a gradient of acetonitrile (8% to 32%) in 10 mM ammonium bicarbonate (pH 9.0) over 60 min into 60 fractions. The peptides were then combined into 18 fractions and dried by vacuum centrifugation.

Peptides from each fraction were dissolved in 0.1% formic acid, directly loaded onto a reversed-phase analytical column (Thermo Scientific EASY-Column, 10 cm, ID 75 µm, 3 µm, C18-A2) using an EASY-nLC 1000 UPLC system (Thermo Fisher Scientific, San Jose, USA). The gradient was comprised of an increase from 7% to 25% solvent B (0.1% FA in 90% acetonitrile) over 26 min, another increase 25% to 40% in 8 min, a further increase to 80% in 3 min, and then holding at 80% for the last 3 min at a constant flow rate of 300 nL/min. The peptides were subsequently analyzed via tandem mass spectrometry (MS/MS) using a Q Exactive™ Plus (Thermo Fisher Scientific). The applied electrospray voltage was 2.0 kV. Intact peptides were detected in the orbitrap at a resolution of 70,000 and ion fragments were detected in the orbitrap at a resolution of 17,500. In the MS survey scan, a

data-dependent mode with automatic alteration (1 MS scan followed by 20 MS/MS scans) was used for the top 20 precursor ions above a threshold ion count of $5 \times 10^4$ with 30 s exclusion. Automatic gain control (AGC) was used to prevent overfilling of the orbitrap; $5 \times 10^4$ ions were accumulated for generating MS/MS spectra. For MS scans, the m/z scan was 350–1800 and the fixed first mass was set at 100 m/z.

The resulting MS/MS data were processed using Mascot 2.2 and Proteome Discoverer 1.4. The mass tolerance for precursor ions was set to 20 ppm in the first search and 5 ppm in the main search, and that for fragment ions was set to 0.02 Da. The false discovery rate (FDR) was adjusted to <1% at the protein, peptide, and PSM (peptide spectrum match) levels, and the minimum score for peptides was set >40. Only unique peptides were used for the protein quantification. Additionally, we have included the computational results of our proteomics study in the supplementary materials: Supplementary Material 1 Protein Identification List; Supplementary Material 2 Peptide Identification List; Supplementary Material 3 Protein Quantification and Differential Analysis List.

Gene Ontology (GO) annotation was derived from the UniProt-GOA database (http://www.ebi.ac.uk/GOA/). First, the identified protein ID was converted to a UniProt ID and then mapped to the GO ID. If some identified proteins were not annotated by the UniProt-GOA database, InterProScan software (a sequence analysis application) was used to annotate the protein's GO function based on the protein sequence alignment method. The DEPs were then classified by GO annotation based on three categories (GO term level 1): biological process, cellular component, and molecular function. The Clusters of Orthologous Groups (COG) of the protein database were used for the functional classification of DEPs.

### Correlation between metabolome and the intestinal microbiome

First, the relative abundance of 23 bacterial groups with significant differences at genus level (LEfSe LDA > 2.0 and $P < 0.05$) obtained by 16S rRNA gene amplicon sequencing and the abundance of 27 bile acids and 62 metabolites obtained by metabolomics analysis were used for analysis. Considering the non-normal distribution of the original data, the Spearman analysis was used to calculate the correlation coefficient. Based on the matched data from the same individual including the abundance of bacterial genera and the plasma levels of identified BAs or metabolites, hierarchical clustering of Spearman correlations was performed with the software R 3.4.2 Heatmap package. To plot a heatmap with correlation coefficient, r is shown in various degree of red and blue indicating positive and negative relationship, respectively.

### Statistics and reproducibility

The analysis between two groups was performed using a two-tailed Student's $t$ test or Mann–Whitney U test, while analysis involving three or more groups was conducted using one-way ANOVA and Kruskal–Wallis. Statistical significance was set at $P \leq 0.05$.

### Reporting summary

Further information on research design is available in the Nature Portfolio Reporting Summary linked to this article.

### Data availability

Primary experimental data will be shared upon personal request by the corresponding authors. 16S rRNA gene sequencing data were deposited in the National Library of Medicine National Center for Biotechnology Information (Accession numbers are PRJNA1032695 and PRJNA1087524), and are publicly available. The mass spectrometry proteomics data have been deposited to the ProteomeXchange Consortium (http://proteomecentral.proteomexchange.org) *via* the iProX partner repository[47,48] with the dataset identifier PXD048019. The source data underlying Figs. 4g, 6a, and 6b can be found in Supplementary data 1, 2, and 3.

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

## Acknowledgements
Authors wish to thank the Zhongshan Hospital of Chinese Medicine and the animal caretakers for support of animal studies. Authors also acknowledge the Centre of Basic Integrative Medicine, School of Basic Medical Sciences of GZUCM for the support of facilities. Authors also want to thank the National Natural Science Foundation of China (82174244, 81973720), Guangzhou University of Chinese Medicine (2021xk40), Key-Area Research and Development Program of Guangdong Province (2020B1111100010), and Social Science and Technology Development Program of Dongguan (No.2020507150119175, No. 20221800905402).

## Author contributions
Huanhuan Luo, Yongliang Zhang and Jianwen Guo conceived and designed the experiments. Song Chen, Zongren Hu and Jianbang Tang wrote the manuscript. Haipeng Zhu, Yuhua Zheng, Jiedong Xiao and Youhua Xu conducted experiments. Yao Wang, Yi Luo, Xiaoying Mo and Yalan Wu analyzed the data. All authors reviewed the manuscript.

## Competing interests
The authors declare no competing interests.
