## [Peer review file · Communications Biology]

Reviewers' comments:

Reviewer #1 (Remarks to the Author):

In the current manuscript, the authors conducted a series of studies in mice to investigate how environmental temperature and humidity affect lipid metabolism, bile acids, microbiota and GLP-1 secretion. This is a very interesting topic since bile acid, microbiota are certainly associated with the temperature and humidity of the environment. GLP-1, as an important hormone from the gut, has a number of functions in metabolic homeostasis. They found that high temperature-and-humidity is linked to abnormal bile acids, microbiota dysbiosis, lower GLP-1 level and disturbed lipid metabolism.

The manuscript is relatively well written. However, I do have some suggestions in terms of the study design:

1. In the first part, HH and HTH environment suppressed GLP-1 secretion. I assume this is the fasting GLP-1 level. What about the postprandial GLP-1 response, since GLP-1 plays its role predominantly during the postprandial phase? Did the authors conduct OGTT or other meal tests? Same issue with other studies - only fasting GLP-1 was measured.
2. What is the effect of HH and HTH on food consumption?

Reviewer #2 (Remarks to the Author):

The manuscript 'Environmental high-temperature-and-humidity disturbs lipid metabolism, bile acids-gut microbiota interaction, and glucagon-like peptide-1 secretion' describes the impact of high temperature housing conditions on mouse intestinal physiology and metabolism. The topic is an interesting and relevant one, as the direct effects of high temperature on host metabolism and microbiome in mammals are not well studied. Further, the experimental approach is generally rigorous and the results are clearly described. Interestingly, the authors find increased oxygenation of the gut with heat exposure, and they nicely performed several interventions, including antibiotics treatment and butyrate and bile acid supplementation, to validate some of their findings and to show an effect of bile acids on heat-induced hormonal changes. However, there are several important points, especially regarding the under-powered omics analyses, which dampen this reviewer's enthusiasm for the paper.

1. Heat exposure is known to suppress appetite, and many of the metabolic changes described (e.g. decrease in appetite-regulating hormones, bile acids) might be a result of reduced food intake. Was food intake of the mice measured?
2. The sample size for microbiome sequencing, and for the omics analyses in general, is small. Are the sequencing data (Fig 2) and metabolomics/proteomics data (Fig 5-6) each from one cage of mice per group? The authors should be careful about over-interpreting possible cage effects as real changes between the groups. This concern is re-enforced by the fact that the two microbiome sequencing runs (Fig 2 and Fig S5) do not seem to come up with the same hits, nor do time points within the groups. If there were real, stable microbiome changes by heat exposure, you would hope for some consistent hits, regardless of whether the timepoints of analysis are slightly different.

Although the efforts to tie the two experiments together in Fig 7C-D is commendable, the link seems tenuous at best as it does not pass even an unadjusted p-value test of 0.05.

3. Related to the concerns on power above in comment 2: although it appears from the methods that FDR correction of microbiome analysis was indeed performed, it should be clarified also in the figure legends or main text whenever the p-values are FDR-adjusted. Was this performed in all analyses selecting differentially abundant taxa, KEGG pathways, proteins and metabolites? (LefSE for example is by default extremely permissive and does not FDR correct).

4. The KEGG pathways yielded by the Picrust analysis are nonsensical as they are all eukaryotic KEGG pathways, whereas the analysis was performed on microbiome 16S sequencing. Are the authors certain that this analysis was performed correctly? If so the results should either be filtered to only bacterially-annotated KEGG pathways or should be excluded from the paper as it provides no meaningful information. For the same reasons as stated above (small sample size, cage effects) the results are in any case difficult to interpret with confidence.

5. There are several instances in the text where the authors over-state or over-interpret their findings. E.g. abstract line 11 or conclusions line 485; there is no evidence presented that Ruminococcaceae and Parabacteroides “regulate” or causally affect any metabolites, it is correlation at best. To support these statements there would need to be some intervention showing that mice supplemented with those bacteria had altered bile acid levels and metabolism.

6. It is interesting that oxygenation of the intestine increased with heat exposure. Oxygenation of the gut is frequently a marker of inflammation and intestinal dysbiosis. The group previously reported increased inflammation in such mice after 8 weeks. Were any such inflammatory markers also already elevated at 2 weeks?

Minor comments

- line 328 LDA cutoff missing from text
- line 119, 329 and all examples: 16S rDNA  16S rRNA gene
- It is not clear in Fig S5 what the inset panel is for
- Line 390-392; Prevotellaceae are not known butyrate producers, butyrate production is typically associated with members of the Firmicutes phylum

Reviewer #3 (Remarks to the Author):

This manuscript described a study exploring how environmental high temperature and humidity impact on intestinal homeostasis. While this is an interesting and timely report, given the changes brought about by global heating, with copious data sets, there are issues that need to be addressed prior to it being ready for publication (see detailed comments below).

Major:

1. The volume of data provides interesting insights into individual aspects of environmental change on microbiome, bile acid profiles, intestinal and liver physiology. However, the compilation of the data do not demonstrate a coherent, well-justified set of experiments. They are somewhat observational and lack physiological insight into how one observation is linked to another.
2. No thought has been given to activation of the stress axis induced by the climatic conditions.

Introduction

1. The statement ‘One potential supporting indication is the association between the increase in global temperatures and the elevated prevalence of diabetes’ is highly speculative and doesn’t provide any context on the more accepted theories relating to the rise in incidence of diabetes (also, specify which type).

2. Similarly, the statement 'Therefore, it's reasonable to deduce that the combination of these two environmental factors, high temperatures and humidity (HTH), would exert a profound impact on gut microbiota and probably contribute to the pathogenesis of obesity, diabetes, and other climate-associated metabolic-inflammatory diseases.' is highly speculative. More specific detail on the mechanisms proposed in the cited literature needs to be presented to contextualise the statement and indeed the study hypothesis
3. Same comment on the data from South China – all very circumstantial and likely to be multifactorial
4. Clearly state the study aims/goals.

Results

1. State acronyms on first mention and use only acronyms after that.
2. Use of fluorescence to measure markers of hypoxia may not provide accurate measurements, and may contribute to the aberrant results between the liver and the gut. Also, explain why hypoxia in the liver might modify release of GLP-1 from L-cells?
3. 'no remarkable clinical manifestation, except for the intermittent diarrhea' – incidences of diarrhoea are physiologically and clinically notable – please provide further details.
4. The inclusion of ob/ob mice in this study has not been justified – obesity and metabolic dysfunction in these mice make it very difficult to draw conclusions about an environmental change. More detail is needed – what induced the weight gain, was food consumption measured? Were metabolic cages used?
5. Some of the data in the supplemental data should be included in the main paper – changes in pancreatic beta cells is fundamental to justifying altered GLP-1 secretion. However, these conclusions are too big a leap – detailed experimental work needs to link all of the experiments conducted.
6. The description of changes in BAs is very dense – please simplify to allow readers to draw conclusions.
7. Explain why CDCA and LCA were selected for in vitro experiments on NCL-716 cells.
8. It would be important, through the use of selective inhibitors, to confirm the role of FXR, rather than TGR5 as the mechanisms by which HTH modified BA metabolism and GLP-1 secretion.
9. In the transomic analysis, why was the focus on LDA/TLCA, given the finding that FXR was the likely receptor important for changes induced by HTH?

Discussion

1. As noted in the introduction, conclusions must be supported by direct evidence – the statement 'our study sheds new light on the pathogenesis of several ambient HTH-associated diseases, like diabetes, enteritis, and probably colorectal cancer,' – the findings do not use models of any of these diseases and so cannot support this claim.

Detailed Response to Reviewers

We would like to express our sincere gratitude to the reviewers for their thoughtful and constructive feedback on our manuscript, which greatly help us improve the quality of our work. We have carefully addressed all their concerns and suggestions in the revised version of the manuscript. They are described in detail in the following point-to-point session. We hope the manuscript can be considered for publication in its revised version.

Comments from the Reviewers:

Reviewer #1:

In the current manuscript, the authors conducted a series of studies in mice to investigate how environmental temperature and humidity affect lipid metabolism, bile acids, microbiota and GLP-1 secretion. This is a very interesting topic since bile acid, microbiota are certainly associated with the temperature and humidity of the environment. GLP-1, as an important hormone from the gut, has a number of functions in metabolic homeostasis. They found that high temperature-and-humidity is linked to abnormal bile acids, microbiota dysbiosis, lower GLP-1 level and disturbed lipid metabolism. The manuscript is relatively well written. However, I do have some suggestions in terms of the study design:

Reviewer point #1:

1. In the first part, HH and HTH environment suppressed GLP-1 secretion. I assume this is the fasting GLP-1 level. What about the postprandial GLP-1 response, since GLP-1 plays its role predominantly during the postprandial phase? Did the authors conduct OGTT or other meal tests? Same issue with other studies - only fasting GLP-1 was measured.

Author response #1:

We appreciate the reviewer's insightful feedback and have carefully revised the manuscript accordingly. Since the suppression of GLP-1 in the HTH group was more pronounced than in the HH group, we focused on the HTH treatment for in-depth investigation of the GLP-1 suppression mechanisms. We measured postprandial GLP-1, with the results presented in **Fig. 1d**, and are also detailed as follows. The results

indicated that HTH conditions can also lead to a reduction in postprandial GLP-1, with a significant decrease observed on day 14. We have described the results of postprandial GLP-1 in the **Results** section.

Page 4, Line 130-132

Results: We measured the postprandial GLP-1 levels, and the results indicated that HTH conditions could lead to a reduction in postprandial GLP-1, with a significant decrease observed on day 14 (Fig. 1d).

Figure 1. The impact of environmental HH and HTH on bio-clinical parameters in mice.

(a) Plasma levels of fasting GLP-1 (b) ghrelin, (c) insulin, (d) postprandial GLP-1. (e) Food intake amount of Day7. $n=6$ for each group. $*P<0.05$, $**P<0.01$ by two-tailed Student's t-test. Error bars represent \pm s.d. (f) Food intake amount of Day14. $n=14$ for each group. $*P<0.05$ by two-tailed Student's t-test. Error bars represent \pm s.d.

Thank you for your comment regarding the Oral Glucose Tolerance Test (OGTT). We have conducted the OGTT, and the results demonstrate that HTH conditions worsened glucose tolerance in mice. The blood glucose levels at all time points after glucose administration were significantly lower in the HTH group compared to control mice (Supplementary Figure2. i). This finding was further supported by higher Area Under the Curve (AUC) values in the HTH group compared to control mice (Supplementary Figure2. j).

We have described the results of OGTT in the **Results** section.

Page 4, Line 122-125

Results: The blood glucose levels at all time points after glucose administration were significantly lower in the HTH group compared to control mice (Supplementary Figure2. i). This finding was further supported by higher Area Under the Curve (AUC) values in the HTH group compared to control mice (Supplementary Figure2. J).

Figure S2. Phenotypic parameters after high-temperature and humidity (HTH) exposure in BABL/c mice.

(a) Body weight (*mean*±*s.e.m*) (b) Plasma fasting glucose levels. (c) Plasma peptide YY (PYY) levels. (d) Plasma gastric inhibitory polypeptide (GIP) levels. (e) Plasma triglyceride levels. (f) Plasma high-density lipoprotein (HDL) levels. (g) Plasma low-density lipoprotein (LDL) levels. (h) Plasma total cholesterol levels. The difference between HTH and normal control (NC) groups was not significant. (i) and (j)OGTT.

Reviewer point #2:

What is the effect of HH and HTH on food consumption?

Author response #2:

We thank the reviewer for their suggestion and acknowledge the importance of food consumption. The relevant results are presented in Fig. 1e and f, as well as detailed below. The findings indicate that HTH conditions can lead to a decrease in food consumption, with a significant reduction observed on day 14. We have detailed the results of food consumption in the **Results** section of the manuscript.

Page 4, Line 132-134

Results: Additionally, HTH conditions were found to cause a reduction in food intake, which became more pronounced on day 14 (Fig. 1e and f).

Figure 1. The impact of environmental HH and HTH on bio-clinical parameters in mice.

(a) Plasma levels of fasting GLP-1 (b) ghrelin, (c) insulin, (d) postprandial GLP-1. (e) Food intake amount of Day7. $n=6$ for each group. $*P<0.05$, $**P<0.01$ by two-tailed Student's t-test. Error bars represent \pm s.d. (f) Food intake amount of Day14. $n=14$ for each group. $*P<0.05$ by two-tailed Student's t-test. Error bars represent \pm s.d.

Reviewer #2:

The manuscript 'Environmental high-temperature-and-humidity disturbs lipid metabolism, bile acids-gut microbiota interaction, and glucagon-like peptide-1 secretion' describes the impact of high temperature housing conditions on mouse intestinal physiology and metabolism. The topic is an interesting and relevant one, as the direct effects of high temperature on host metabolism and microbiome in mammals are not well studied. Further, the experimental approach is generally rigorous and the results are clearly described. Interestingly, the authors find increased oxygenation of the gut with heat exposure, and they nicely performed several interventions, including antibiotics treatment and butyrate and bile acid supplementation, to validate some of their findings and to show an effect of bile acids on heat-induced hormonal changes. However, there are several important points, especially regarding the under-powered omics analyses, which dampen this reviewer's enthusiasm for the paper.

Reviewer point #1:

1.Heat exposure is known to suppress appetite, and many of the metabolic changes described (e.g. decrease in appetite-regulating hormones, bile acids) might be a result of reduced food intake. Was food intake of the mice measured?

Author response #1:

We thank the reviewer for their suggestion and acknowledge the importance of food intake. The relevant results are presented in Fig. 1e and f, as well as detailed below. The findings indicate that HTH conditions can lead to a decrease in food intake, with a significant reduction observed on day 14. We have detailed the results of food intake in the **Results** section of the manuscript.

Page 4, Line 132-134

Results: Additionally, HTH conditions were found to cause a reduction in food intake, which became more pronounced on day 14 (Fig. 1e and f).

Figure 1. The impact of environmental HH and HTH on bio-clinical parameters in mice.

(a) Plasma levels of fasting GLP-1 (b) ghrelin, (c) insulin, (d) postprandial GLP-1. (e) Food intake amount of Day7. $n=6$ for each group. * $P < 0.05$, ** $P < 0.01$ by two-tailed Student's t-test. Error bars represent \pm s.d. (f) Food intake amount of Day14. $n=14$ for each group. * $P < 0.05$ by two-tailed Student's t-test. Error bars represent \pm s.d.

Reviewer point #2:

2.The sample size for microbiome sequencing, and for the omics analyses in general, is small. Are the sequencing data (Fig 2) and metabolomics/proteomics data (Fig 5-6) each from one cage of mice per group? The authors should be careful about over-interpreting possible cage effects as real changes between the

groups. This concern is re-enforced by the fact that the two microbiome sequencing runs (Fig 2 and Supplementary Figure5) do not seem to come up with the same hits, nor do time points within the groups. If there were real, stable microbiome changes by heat exposure, you would hope for some consistent hits, regardless of whether the timepoints of analysis are slightly different. Although the efforts to tie the two experiments together in Fig 7C-D is commendable, the link seems tenuous at best as it does not pass even an unadjusted p-value test of 0.05.

Author response #2:

I apologize for any confusion caused by my previous explanation. To clarify, the 16S rRNA gene sequence presented in Fig. 2 originate from Experiment 1, as shown in Supplementary Figure1. The proteomics data in Fig. 5 and the metabolomics data in Fig. 6 are derived from Experiment 3, also indicated in Supplementary Figure1. Due to the high cost of proteomics analysis, and considering that other studies have also conducted experiments with three mice¹, we chose to test three mice in the proteomics experiment shown in Fig. 5.

The independent samples t-test, which we have employed in our proteomics analysis, is a robust statistical method used to compare the means of two independent groups. The use of three subjects per group in such experiments is a common practice and is considered acceptable in the field of biomedical research. While larger sample sizes generally provide greater power, the independent samples t-test is known for its efficiency and can yield reliable results even with smaller sample sizes².

Additionally, we have included the computational results of our proteomics study in the supplementary materials: Supplementary Material 1: Protein Identification List; Supplementary Material 2: Peptide Identification List; Supplementary Material 3: Protein Quantification and Differential Analysis List.

Thank you for your query about the discrepancy between the two microbiome sequencing runs (Fig 2 and Supplementary Figure5). As you correctly noted, we have removed Supplementary Figure5 from the manuscript.

We also appreciate your concern regarding Fig 7C-D. We agree that the association does not meet the unadjusted p -value threshold of 0.05. Therefore, we have excluded these results from our revised manuscript.

Figure S1. Schematic diagram of the experiment design.

NC: normal control group. HH: high humidity. HTH: high temperature and high humidity.

Reviewer point #3:

3.Related to the concerns on power above in comment 2: although it appears from the methods that FDR correction of microbiome analysis was indeed performed, it should be clarified also in the figure legends or main text whenever the p -values are FDR-adjusted. Was this performed in all analyses selecting differentially abundant taxa, KEGG pathways, proteins and metabolites? (LefSE for example is by default extremely permissive and does not FDR correct).

Author response #3:

Thank you for your valuable feedback and for pointing out the necessity of clarity regarding the False Discovery Rate (FDR) correction in our manuscript. In response to your concern, we have taken the following steps to ensure the rigor and clarity of our statistical analysis:

Removal of Unadjusted Data: We have carefully reviewed our analyses and have removed data that were not subjected to FDR correction. This includes data presented in Fig 2 (I and J) and Fig 7 (C and D), which were identified as not having undergone FDR adjustment.

FDR Correction Application: For all remaining analyses involving differentially abundant taxa, proteins, and metabolites, we have ensured that the FDR correction has

been applied. This includes meticulous scrutiny of our statistical methods to confirm the application of FDR adjustments across all relevant datasets.

Clarification in Figure Legends and Text: To further enhance clarity, we have updated all figure legends. We have explicitly stated that "Statistical significance was evaluated using p -values adjusted for multiple comparisons with the FDR method." This statement has been incorporated to clearly indicate where FDR-adjusted p -values have been used, thus providing the reader with a transparent and accurate understanding of our statistical approach. We believe these revisions address your concerns effectively and enhance the scientific rigor of our manuscript.

Reviewer point #4:

4. The KEGG pathways yielded by the Picrust analysis are nonsensical as they are all eukaryotic KEGG pathways, whereas the analysis was performed on microbiome 16S sequencing. Are the authors certain that this analysis was performed correctly? If so the results should either be filtered to only bacterially-annotated KEGG pathways or should be excluded from the paper as it provides no meaningful information. For the same reasons as stated above (small sample size, cage effects) the results are in any case difficult to interpret with confidence.

Author response #4:

Thank you for your critical assessment of our manuscript, particularly concerning the KEGG pathway analysis results derived from the Picrust analysis of microbiome 16S rRNA gene sequencing data. Your observation about the identification of predominantly eukaryotic pathways in a microbial context is indeed crucial and well-noted.

Upon revisiting our analysis, we have decided to remove the results pertaining to the KEGG pathway analysis from our paper. We agree that the inclusion of this data, without clear bacterial annotations, could potentially lead to misinterpretation and does not add meaningful insight to our research findings. We believe that excluding this part of the analysis will strengthen the overall quality and focus of our manuscript.

Reviewer point #5:

5. There are several instances in the text where the authors over-state or over-interpret their findings. E.g. abstract line 11 or conclusions line 485; there is no evidence presented that Ruminococcaceae and Parabacteroides “regulate” or causally affect any metabolites, it is correlation at best. To support these statements there would need to be some intervention showing that mice supplemented with those bacteria had altered bile acid levels and metabolism.

Author response #5:

Thank you for your insightful feedback on our manuscript. We appreciate your attention to the precision of our language and the accuracy of the interpretations of our findings. In light of your comments, particularly regarding the sections in the abstract (line 11) and the conclusions (line 485), we have thoroughly reviewed our manuscript and have taken immediate action. We acknowledge that the previous wording suggested a causal relationship between *Ruminococcaceae*, *Parabacteroides*, and specific metabolites, which our current data does not conclusively support. As you correctly pointed out, our findings at this stage indicate correlation rather than causation. To address this, we have removed the statements that implied direct regulation or causal effects by *Ruminococcaceae* and *Parabacteroides* on metabolites. Your feedback has been instrumental in helping us improve the clarity and scientific integrity of our work.

Reviewer point #6:

It is interesting that oxygenation of the intestine increased with heat exposure. Oxygenation of the gut is frequently a marker of inflammation and intestinal dysbiosis. The group previously reported increased inflammation in such mice after 8 weeks. Were any such inflammatory markers also already elevated at 2 weeks?

Author response #6:

Thank you for pointing out the intriguing aspect of increased intestinal oxygenation with heat exposure in our study. In response to your query regarding the presence of inflammatory markers, we conducted additional experiments. We employed PCR

techniques to assess the expression of inflammatory markers such as IL-6, MCP-1, and TNF- α in the colon. As illustrated in the attached figure, our results indicate a significant elevation in the expression levels of IL-6, MCP-1, and TNF- α after 2 weeks of exposure to HTH conditions.

We hope this additional data addresses your question and thank you for your insightful comment.

Figure: The mRNA expression of IL-6, MCP-1, TNF- α in colon. $n=6$ mice for each group. * $P<0.05$, ** $P<0.01$, *** $P<0.001$ by two-tailed Student's t-test. Error bars represent \pm s.d.

Minor comments

- *line 328 LDA cutoff missing from text*

Author response: Thank you for pointing out the omission regarding the Linear Discriminant Analysis (LDA) cutoff in our manuscript. Your attention to detail is greatly appreciated and has helped us improve the clarity and completeness of our presentation. In response to your comment, we have revised the manuscript and have now included the specific LDA cutoff value in the text. The LDA score threshold used in our analysis was $LDA > 2.0$. This information has been added to the revised manuscript at line 297, ensuring that readers are fully informed about the criteria used for our analysis.

• ***line 119, 329 and all examples: 16S rDNA  16S rRNA gene***

Author response: Thank you for your careful review of our manuscript and for your specific suggestion regarding the terminology of the 16S ribosomal RNA gene.

In accordance with your recommendation, we have revised our manuscript to ensure accurate and consistent terminology. We have replaced all instances of "16S rDNA" with "16S rRNA gene" throughout the text. We are grateful for your guidance in this matter. This correction not only aligns our manuscript with standard nomenclature but also enhances the clarity and accuracy of the information presented.

• ***It is not clear in Supplementary Figure5 what the inset panel is for***

Author response: Thank you for your feedback and for pointing out the lack of clarity regarding Supplementary Figure5 in our manuscript. In response to your comment, we have carefully reviewed the figure and have made the decision to remove Supplementary Figure5 from the manuscript. We realized that the inset panel in question did not contribute significantly to the overall understanding or interpretation of our findings. We believe that the removal of this figure simplifies the presentation and focuses attention on the more relevant and crucial data within our study.

We are committed to ensuring that our manuscript is as clear and concise as possible, and we appreciate your help in identifying areas where we can improve. Your insightful feedback has been invaluable in enhancing the overall quality of our work.

• ***Line 390-392; Prevotellaceae are not known butyrate producers, butyrate production is typically associated with members of the Firmicutes phylum***

Author Response: Thank you for pointing out the inaccuracy in our description of the *Prevotellaceae* family's relationship to butyrate production, as referenced in lines 390-392 of our manuscript. We acknowledge and agree with your observation that butyrate production is more typically associated with members of the Firmicutes phylum, rather than *Prevotellaceae*. Consequently, we have revised the text in lines 340-341 on page 8 of the Results section to read: "*Firmicutes* probably contributed to the decreased plasma butyric acid level observed in our study."

Reviewer #3:

This manuscript described a study exploring how environmental high temperature and humidity impact on intestinal homeostasis. While this is an interesting and timely report, given the changes brought about by global heating, with copious data sets, there are issues that need to be addressed prior to it being ready for publication (see detailed comments below).

Major:

Reviewer point #1:

1. The volume of data provides interesting insights into individual aspects of environmental change on microbiome, bile acid profiles, intestinal and liver physiology. However, the compilation of the data do not demonstrate a coherent, well-justified set of experiments. They are somewhat observational and lack physiological insight into how one observation is linked to another.

Author response #1:

Thank you for your insightful feedback. Following your advice, we have incorporated additional connective language in each section of our results to better link our findings and provide a coherent narrative. For instance, on page 4, lines 142-146, we state: "The intestinal foci, where L-cells inhabit, should be profoundly impacted by HTH and that plays a pivotal role in HTH-induced abnormalities. Therefore, we first investigate the impact of HTH treatment on gut dysbiosis with 16S rRNA gene sequencing of fecal samples." This statement serves to introduce the topic of gut microbiota in the context of HTH.

On page 5, lines 174-178, we elaborate: "Gut microbiota-derived SCFAs act as signaling molecules stimulating GLP-1 secretion by binding to their transmembrane receptors FFAR2 (GPR43) and FFAR3 (GPR41) on colonic L-cell response³.

Moreover, SCFAs modulate excitability and proliferation of enteric neurons^{4,5} which play critical role in GLP-1 regulation." This section links gut microbiota with SCFAs, leading into a discussion about SCFAs.

Further, on page 5, lines 193-196, we discuss: "The gut microbiota plays a significant role in the synthesis of bile acids⁶. For example, *Alistipes* can produce succinic acid⁷. We speculated that the alteration of BAs transformed by gut microbiota^{8,9}, should be

responsible for the decreased GLP-1 in HTH group." This part connects gut microbiota with bile acids, introducing the topic of bile acids.

Finally, on page 6, lines 220-224, we add: "Bile acids are well-recognized stimuli of GLP-1 secretion, activation of FXR decreases proglucagon expression by interfering with the glucose-responsive factor Carbohydrate-Responsive Element Binding Protein (ChREBP) and GLP-1 secretion by inhibiting glycolysis¹⁰. Whether FXR signaling pathway plays a dominated role in HTH-induced GLP-1 suppression mediated by aberrant BAs remains uncertain." This section links GLP-1 with the FXR signaling pathway.

These additions aim to address your concerns. We hope these revisions enhance the coherence of our manuscript.

Reviewer point #2:

2. No thought has been given to activation of the stress axis induced by the climatic conditions.

Author response #2:

Thank you for your insightful comment. We agree with you that climatic changes can lead to alterations in the stress axis. However, our study specifically focuses on the impact of HTH conditions on mice, examining how the induced stress under these conditions can lead to a decline in GLP-1. We appreciate your feedback and will consider it for future studies that might explore broader aspects of environmental stress.

Introduction :

Reviewer point #1:

1. The statement ‘One potential supporting indication is the association between the increase in global temperatures and the elevated prevalence of diabetes’ is highly speculative and doesn’t provide any context on the more accepted theories relating to the rise in incidence of diabetes (also, specify which type).

Author response #1:

Thank you for your critical assessment of our manuscript, specifically regarding the statement about the potential association between the increase in global temperatures

and the prevalence of diabetes. We appreciate your attention to the accuracy and scientific grounding of our discussions. In light of your feedback, we have re-evaluated the mentioned statement and recognize that it was indeed speculative and lacked the necessary context and support from more accepted theories regarding the rise in the incidence of diabetes. As a result, we have removed this statement from our manuscript.

We are grateful for your insightful comments, which have guided us in improving the quality and reliability of our work.

Reviewer point #2:

2. Similarly, the statement ‘Therefore, it’s reasonable to deduce that the combination of these two environmental factors, high temperatures and humidity (HTH), would exert a profound impact on gut microbiota and probably contribute to the pathogenesis of obesity, diabetes, and other climate-associated metabolic-inflammatory diseases.’ is highly speculative. More specific detail on the mechanisms proposed in the cited literature needs to be presented to contextualise the statement and indeed the study hypothesis

Author response #2:

In response to the reviewer's concern regarding the speculative nature of our statement about the combined effects of HTH on gut microbiota and their potential contribution to the pathogenesis of obesity, diabetes, and other climate-associated metabolic-inflammatory diseases, we have taken your feedback into consideration. Acknowledging that the statement may have been overly speculative without sufficient supporting details on the proposed mechanisms, we have removed this statement from our manuscript. We appreciate your guidance in ensuring the accuracy and specificity of our study.

Reviewer point #3:

3. Same comment on the data from South China – all very circumstantial and likely to be multifactorial

Author response #3:

Thank you for your valuable feedback on our manuscript, particularly regarding the section discussing the data from South China. We appreciate your emphasis on the importance of presenting well-substantiated and comprehensive analysis in our research. Taking your comment into consideration, we have made the decision to remove this section from our manuscript. Thank you once again for your thorough review and constructive suggestions, which are invaluable to the improvement of our work.

Reviewer point #4:

4. Clearly state the study aims/goals.

Author response #4:

This investigation is a continuation of our previous work, where we observed changes in the gut microbiota under HTH conditions. Building on our prior findings, this study further reveals the reduced LCA levels can suppress GLP-1 through the activation of the FXR. By elucidating these mechanisms, we aim to contribute valuable insights into how climatic conditions like HTH may influence metabolic diseases, advancing our understanding of the interplay between environment, gut microbiota, and human health. We have added the aims in our **Introduction** section.

We hope that these revisions adequately address your point and clarify the aims and goals of our study, thus enhancing the overall coherence and impact of our research.

Page 3, Line 105-109

Building on our prior findings, this study further reveals the reduced lithocholic acid (LCA) levels can suppress GLP-1 through the activation of the farnesoid X receptor (FXR). By elucidating these mechanisms, we aim to contribute valuable insights into how climatic conditions like HTH may influence metabolic diseases, advancing our understanding of the interplay between environment, gut microbiota, and human health.

Results :

Reviewer point #1:

1.State acronyms on first mention and use only acronyms after that.

Author response #1:

Thank you for your valuable feedback regarding the use of acronyms in our manuscript. We appreciate your guidance on ensuring clarity and consistency in our presentation.

In accordance with your suggestion, we have carefully revised our manuscript to ensure that all acronyms are fully spelled out at their first mention. Following these initial introductions, we have consistently used only the acronyms throughout the rest of the text. This revision has been applied to the entire manuscript to ensure uniformity and to aid in the readability and comprehension of our work.

We agree that this approach improves the clarity of the manuscript and makes it more accessible to readers, especially in sections where multiple acronyms are used. We are grateful for the opportunity to enhance the quality of our presentation based on your insightful feedback.

Thank you once again for your thorough review and helpful suggestions.

Reviewer point #2:

2. Use of fluorescence to measure markers of hypoxia may not provide accurate measurements, and may contribute to the aberrant results between the liver and the gut. Also, explain why hypoxia in the liver might modify release of GLP-1 from L-cells?

Author response #2:

Thank you for your insightful comment concerning the use of fluorescence to measure markers of hypoxia in our study. We greatly value your expertise and have taken your feedback into serious consideration.

Upon re-evaluating our methodology and findings, we have indeed recognized that the relationship between hypoxia and GLP-1 in our study was not as clear-cut as initially presumed. In light of this, and considering your concerns about the accuracy of fluorescence measurements for hypoxia markers, we have decided to remove the results related to hypoxia from our manuscript.

We believe that this decision enhances the scientific rigor of our work by focusing on data and results that are more robust and directly relevant to the core objectives of our

study. We are committed to ensuring the highest standards of research integrity and accuracy.

Thank you again for your thorough review and constructive feedback, which have been instrumental in refining our manuscript.

3. *'no remarkable clinical manifestation, except for the intermittent diarrhea' – incidences of diarrhoea are physiologically and clinically notable – please provide further details.*

Author response #3:

Thank you for your comment highlighting the need for clarity and accuracy in describing clinical manifestations in our study. We appreciate your emphasis on the physiological and clinical significance of symptoms such as diarrhea. In response to your feedback, we have re-evaluated the mentioned section of our manuscript and decided to remove the statement "no remarkable clinical manifestation, except for the intermittent diarrhea" from our manuscript. Thank you once again for your thorough review and valuable feedback, which have significantly contributed to improving the quality of our manuscript.

4. *The inclusion of ob/ob mice in this study has not been justified – obesity and metabolic dysfunction in these mice make it very difficult to draw conclusions about an environmental change. More detail is needed – what induced the weight gain, was food consumption measured? Were metabolic cages used?*

Author response #4:

Thank you for your critical evaluation of our study, particularly regarding the inclusion of *ob/ob* mice. We appreciate your insight into the complexities associated with using this model, especially in the context of assessing environmental changes. Upon reviewing your comments and reevaluating the pertinence of *ob/ob* mice to our study's objectives, we agree that their inclusion was not adequately justified. The inherent obesity and metabolic dysfunctions present in these mice indeed complicate the interpretation of results in relation to environmental changes.

In light of this, we have made the decision to remove the results pertaining to *ob/ob* mice from our manuscript. We recognize that without detailed information on factors

like the cause of weight gain, food consumption, and the use of metabolic cages, the data from these mice could lead to ambiguous or misleading conclusions. We believe this revision strengthens the focus and clarity of our findings, aligning them more closely with our study's primary objectives.

Thank you once again for your thorough review and constructive feedback, which have been invaluable in enhancing the scientific rigor of our work.

5. Some of the data in the supplemental data should be included in the main paper – changes in pancreatic beta cells is fundamental to justifying altered GLP-1 secretion. However, these conclusions are too big a leap – detailed experimental work needs to link all of the experiments conducted.

Author response #5:

Thank you for your valuable suggestion regarding the inclusion of data from our supplementary materials in the main body of our manuscript. In response to your recommendation, we have revised the manuscript accordingly. We agree that Supplementary Image 4 provides critical information that is fundamental to our study. Thus, we have included this image in the main paper, specifically in Figure 2a. The revised Figure 2 now incorporates Supplementary Image 4. And we have described the results in the **Results** section.

Page 4, Line 139-142

Results: As the immunofluorescence staining targeting insulin showed an expanded area of β -cell in HTH treated BABL/c mice (Fig. 2a), indicating an augmented insulin synthesis, we deduced that the decreased plasma insulin should be explained with deficient GLP-1 stimulated secretion.

Figure 2. Gut microbiota induced by environmental HTH.

(a) The pancreatic β cell mass, indicated by insulin immunofluorescence staining, was significantly expanded at 14 days after HTH treatment compared to NC group. The arbitrary units (AU) indicating the ratio of mean gray value to area were shown in columns. Alpha diversity was applied to analyse the complexity of species of the NC group and HTH group. (b) Indices of Simpson and (c) Shannon, as well as (d) OTU numbers, are displayed. (e) PCA at the OTU level. The PCA of all the samples was based on the relative abundance of the OTUs. Percentiles represent the contributions of principal components to differences among samples. (f) The pie plot indicating the composition gut microbiota abundance at phylum level. (g) The ratio of *Firmicutes/Bacteroidetes*. (h) and (i) The gut microbiota with differential abundance at genera level at day 7 and day 14 after HTH treatment. $n=6$ mice for each group. $*P<0.05$, $**P<0.01$ by two-tailed Student's t-test. Error bars represent \pm s.d. Statistical significance was evaluated using p -values adjusted for multiple comparisons with the FDR method.

We acknowledge your concern regarding the need for a more detailed experimental link between the various aspects of our study. We are committed to ensuring that our conclusions are well-supported by comprehensive experimental evidence. As such, we have taken additional steps to strengthen the connections between our experiments and have ensured that these links are clearly articulated in the revised manuscript.

For instance, on page 4, lines 142-146 we state: "The intestinal foci, where L-cells are located, are likely to be profoundly impacted by HTH conditions, playing a pivotal role in HTH-induced abnormalities. Therefore, we first investigate the impact of HTH treatment on gut dysbiosis with 16S rRNA gene sequencing of fecal samples." This statement serves to introduce the topic of gut microbiota in the context of HTH.

On page 5, lines 174-178, we elaborate: "Gut microbiota-derived SCFAs act as signaling molecules stimulating GLP-1 secretion by binding to their transmembrane receptors FFAR2 (GPR43) and FFAR3 (GPR41) on colonic L-cell response³.

Moreover, SCFAs modulate excitability and proliferation of enteric neurons^{4,5} which play critical role in GLP-1 regulation." This section links gut microbiota with SCFAs, leading into a discussion about SCFAs.

Further, on page 5, lines 193-196, we discuss: "The gut microbiota plays a significant role in the synthesis of bile acids⁶. For example, *Alistipes* can produce succinic acid⁷. We speculated that the alteration of BAs transformed by gut microbiota^{8,9}, should be responsible for the decreased GLP-1 in HTH group." This part connects gut microbiota with bile acids, introducing the topic of bile acids.

Finally, on page 6, lines 220-224, we add: "Bile acids are well-recognized stimuli of GLP-1 secretion, activation of FXR decreases proglucagon expression by interfering with the glucose-responsive factor Carbohydrate-Responsive Element Binding Protein (ChREBP) and GLP-1 secretion by inhibiting glycolysis¹⁰. Whether FXR signaling pathway plays a dominated role in HTH-induced GLP-1 suppression mediated by aberrant BAs remains uncertain." This section links GLP-1 with the FXR signaling pathway.

We hope these revisions address your concerns and substantially improve the quality of our manuscript.

6. The description of changes in BAs is very dense – please simplify to allow readers to draw conclusions.

Author response #6:

Thank you for your constructive feedback regarding the density of the bile acids (BAs) section in our manuscript. We understand the importance of presenting our findings in a clear and accessible manner that allows readers to easily draw conclusions. In response to your comments, we have revised the section on BAs to make it more concise and reader-friendly.

Page 5-6, Line 192-217

Results:

Alterations of BAs

The gut microbiota plays a significant role in the synthesis of bile acids⁶. For example, *Alistipes* can produce succinic acid⁷. We speculated that the alteration of BAs transformed by gut microbiota^{8,9}, should be responsible for the decreased GLP-1 in HTH group. We determined the hepatic expression of bile acid synthetic enzymes, including cholesterol 7 α -hydroxylase (CYP7A1), the rate-limiting enzyme in the classic pathway, oxysterol 7 α -hydroxylase (CYP7B1), sterol 12 α -hydroxylase (CYP8B1), the key enzyme for the synthesis of cholic acid (CA), and sterol 27-hydroxylase (CYP27), the initial enzyme in the alternative pathway. As shown in Fig. 4b-4e, the expressions of the enzymes were downregulated, and among them, Cyp7b1 was significantly decreased, indicating a significant suppression in the alternative pathway of BA synthesis, by which chenodeoxycholic acid (CDCA) synthesis was particularly affected.

To further elucidate the impact of BAs on the suppression of GLP-1, we conducted *in vitro* experiments evaluating the effects of CDCA and its derivatives on GLP-1 production (Fig. 4a). Cultured Ncl-716 cells, when treated with chenodeoxycholic acid (CDCA), taurochenodeoxycholic acid (TCDCA), and tauroolithocholic acid (TLCA) at concentrations ranging from 25 to 100 μ M, acted as agonists and exhibited a dose-dependent effect on GLP-1 production. Notably, LCA at a concentration of 100 μ M significantly increased GLP-1 production (Fig. 4f). These findings suggest

that alterations in the composition of BAs may impair GLP-1 production. And then we conducted a targeted metabolomics study using liquid chromatography-mass spectrometry (LC-MS) to assess the differences of hepatic bile acid profile. LCA were decreased in HTH group compared to the NC group (Fig. 4g). These findings suggest that the altered levels of LCA might explain the suppression of GLP-1 induced by HTH, and we choose LCA for further research.

We hope that these revisions address your concerns and improve the clarity of our manuscript. We appreciate your guidance in enhancing the readability and understanding of our study.

Figure 4. BAs homeostasis in mice receiving persistent exposure of environmental HTH.

(a) The diagrammatic sketch of bile acids homeostasis and its regulatory effect on β -oxidation through FXR-ACC2 pathway. The hepatic expression of (b) Cyp7a1, (c) Cyp7b1, (d) Cyp8b1, and (e) Cyp27a1 were shown in columns. n=6 for each group, * P <0.05 by two-tailed Student's t-test. Error bars represent \pm s.d. (f) The level of GLP-1 after treated with LCA, TLCA, CDCA and TCDCa *in vitro*. n=3 for each group, *** P <0.001 by ANOVA and Kruskal-Wallis test. Error bars represent \pm s.d. (g)

Targeted metabolomics analysis of BAs. n=10 for each group, * $P < 0.05$ by two-tailed Student's t-test (LCA) and Mann-Whitney test (TLCA). Error bars represent \pm s.d.

7. Explain why CDCA and LCA were selected for *in vitro* experiments on NCL-716 cells.

Author response #7:

Thank you for your question and we appreciate the opportunity to revise our manuscript.

In our *in vivo* studies, we observed a notable decrease in the expression of Cyp7b1 under HTH conditions. This decrease points to a significant suppression in the alternative pathway of BAs synthesis, which particularly affects the synthesis of CDCA. Given this finding, CDCA was chosen to directly investigate its role in the suppression of GLP-1, as it is a primary bile acid whose synthesis is directly influenced by the observed enzymatic changes.

Additionally, our targeted metabolomics study using liquid chromatography-mass spectrometry (LC-MS) revealed that the levels of LCA were decreased in the HTH group compared to the NC group. LCA, as a secondary bile acid, is a product of microbial metabolism and can have significant physiological effects, including on GLP-1 production. The reduction of LCA in the HTH group suggested its potential role in the observed GLP-1 suppression.

Therefore, the choice of CDCA and LCA for our *in vitro* studies. We hope this explanation provides a clear rationale for our selection.

Figure 4. BAs homeostasis in mice receiving persistent exposure of environmental HTH.

(a) The diagrammatic sketch of bile acids homeostasis and its regulatory effect on β -oxidation through FXR-ACC2 pathway. The hepatic expression of (b) Cyp7a1, (c) Cyp7b1, (d) Cyp8b1, and (e) Cyp27a1 were shown in columns. n=6 for each group, * P <0.05 by two-tailed Student's t-test. Error bars represent \pm s.d. (f) The level of GLP-1 after treated with LCA, TLCA, CDCA and TCDCa *in vitro*. n=3 for each group, *** P <0.001 by ANOVA and Kruskal-Wallis test. Error bars represent \pm s.d. (g) Targeted metabolomics analysis of BAs. n=10 for each group, * P <0.05 by two-tailed Student's t-test (LCA) and Mann-Whitney test (TLCA). Error bars represent \pm s.d.

8. It would be important, through the use of selective inhibitors, to confirm the role of FXR, rather than TGR5 as the mechanisms by which HTH modified BA metabolism and GLP-1 secretion.

Author response #8:

Thank you for suggesting the use of selective inhibitors to discern the specific roles of the FXR and TGR5 in the altered BAs metabolism and GLP-1 secretion under HTH

conditions. We appreciate the importance of this aspect in validating our findings and have accordingly conducted additional experiments to address this point.

As detailed in our manuscript, we have expanded our *in vitro* studies to include the use of FXR inhibitors. Our results showed that, when compared to the negative control (NC) group, the application of FXR inhibitors led to a significant increase in GLP-1 levels. This effect was further pronounced when LCA was combined with FXR inhibitors, suggesting an enhanced GLP-1 production.

These findings provide substantial evidence that LCA promotes GLP-1 production predominantly through the inhibition of FXR. The use of FXR inhibitors allowed us to specifically target and confirm the role of FXR in this process, as shown in Fig. 5d. We believe that these additional experiments and results effectively address your concern and further solidify the mechanisms we propose regarding how HTH conditions modify BAs metabolism and GLP-1 secretion. We are grateful for your suggestion, which guided us to enhance the robustness and clarity of our study's conclusions. We have described the results in the **Results** section.

Page 6, Line 244-248

Results: *In vitro* experimental results revealed that, compared to the negative control (NC) group, FXR inhibitors significantly increased the levels of GLP-1. Additionally, compared to the LCA group, the combination of LCA with FXR inhibitors led to a notable increase in GLP-1 levels. These *in vitro* findings suggest that LCA promotes GLP-1 production through the inhibition of FXR (Fig. 5d).

Figure 5. Ileal proteomics analysis.

(a) Heat map indicates the differential expressed protein (DEP) with a criteria of change folds >1.2 or <0.83 and $p\text{-value} < 0.05$. (b) Volcano plot indicates DEPs with the above criteria and the top 3 enriched proteins and reduced proteins were labeled. (c) Relative abundance of FXR regulated protein. $n=3$ mice for each group. $*P < 0.05$, $**P < 0.01$ by two-tailed Student's t -test. Error bars represent \pm s.d. (d) The level of GLP-1 after treated with LCA, and FXR inhibitor *in vitro*. $n=3$ for each group, $**P < 0.01$ by ANOVA. Error bars represent \pm s.d. Statistical significance was evaluated using p -values adjusted for multiple comparisons with the FDR method.

9. In the transomic analysis, why was the focus on LDA/TLCA, given the finding that FXR was the likely receptor important for changes induced by HTH?

Author response #9:

First and foremost, thank you for your insightful question regarding our transomic analysis. In our transomic approach, we conducted a correlation analysis between bile acids and gut microbiota, rather than directly linking proteomics with gut microbiota. However, as you rightly pointed out, FXR is likely a critical receptor for the changes induced by HTH conditions. Acknowledging this, we have supplemented our study with additional *in vitro* experiments to underscore the importance of FXR. These experiments were designed to further elucidate the role of FXR in mediating the physiological responses to HTH conditions.

Your suggestion has been instrumental in guiding us to enhance the depth of our study, particularly in exploring the mechanistic aspects.

Our results showed that, when compared to the negative control (NC) group, the application of FXR inhibitors led to a significant increase in GLP-1 levels. This effect was further pronounced when LCA was combined with FXR inhibitors, suggesting an enhanced GLP-1 production (Fig. 5d). These findings provide evidence that LCA promotes GLP-1 production predominantly through the inhibition of FXR.

We believe that these additional experiments and results effectively address your concern. We are grateful for your suggestion, which guided us to enhance the robustness and clarity of our study's conclusions. We have described the results in the Results section.

Page 6, Line 244-248

Results: *In vitro* experimental results revealed that, compared to the negative control (NC) group, FXR inhibitors significantly increased the levels of GLP-1. Additionally, compared to the LCA group, the combination of LCA with FXR inhibitors led to a notable increase in GLP-1 levels. These *in vitro* findings suggest that LCA promotes GLP-1 production through the inhibition of FXR (Fig. 5d).

Figure 5. Ileal proteomics analysis.

(a) Heat map indicates the differential expressed protein (DEP) with a criteria of change folds >1.2 or <0.83 and $p\text{-value} < 0.05$. (b) Volcano plot indicates DEPs with the above criteria and the top 3 enriched proteins and reduced proteins were labeled. (c) Relative abundance of FXR regulated protein. $n=3$ mice for each group. $*P < 0.05$, $**P < 0.01$ by two-tailed Student's t -test. Error bars represent \pm s.d. (d) The level of GLP-1 after treated with LCA, and FXR inhibitor *in vitro*. $n=3$ for each group, $**P < 0.01$ by ANOVA. Error bars represent \pm s.d. Statistical significance was evaluated using p -values adjusted for multiple comparisons with the FDR method.

Discussion :

Reviewer point #1:

1. As noted in the introduction, conclusions must be supported by direct evidence – the statement ‘our study sheds new light on the pathogenesis of several ambient HTH-associated diseases, like diabetes, enteritis, and probably colorectal cancer,’ –

the findings do not use models of any of these diseases and so cannot support this claim.

Author response #1:

Thank you for your insightful feedback regarding the claims made in our manuscript. We understand the importance of ensuring that our conclusions are directly supported by the evidence presented. In light of your observation, we have removed the statement ‘our study sheds new light on the pathogenesis of several ambient HTH-associated diseases, like diabetes, enteritis, and probably colorectal cancer’ from our manuscript.

We appreciate your guidance in helping us maintain the scientific rigor and integrity of our work.

I would like to extend my heartfelt thanks for providing me with the opportunity to revise our manuscript. Your detailed and insightful comments have been instrumental in enhancing the overall quality and rigor of our work. The thoroughness of your reviews has not only helped in identifying areas that needed improvement but also in guiding our thought process towards a more comprehensive and clearer presentation of our research.

The opportunity to address your concerns and suggestions has been invaluable. It has allowed us to refine our arguments and clarify our conclusions, thereby contributing significantly to the advancement of our study. I am grateful for the time and effort you have dedicated to reviewing our work, and for the constructive criticism that has pushed us to strive for excellence.

We hope that the revisions we have made are in line with your expectations.

Thank you once again for your invaluable input and for the chance to improve our manuscript.

References

- 1 Barreto, R. *et al.* Mice Placental ECM Components May Provide A Three-Dimensional Placental Microenvironment. *Bioengineering (Basel)* **10**, doi:10.3390/bioengineering10010016 (2022).
- 2 Walpole, R. E., Myers, R. H., Myers, S. L. & Ye, K. E. Probability & statistics for engineers & scientists. *Technometrics* **32** (1998).
- 3 Liu, R. *et al.* Effects of glucagon-like peptide-1 agents on left ventricular function: systematic review and meta-analysis. *Ann Med* **46**, 664-671, doi:10.3109/07853890.2014.949837 (2014).
- 4 Muscogiuri, G. *et al.* GLP-1: benefits beyond pancreas. *J Endocrinol Invest* **37**, 1143-1153, doi:10.1007/s40618-014-0137-y (2014).
- 5 Kim, Y. K., Kim, O. Y. & Song, J. Alleviation of Depression by Glucagon-Like Peptide 1 Through the Regulation of Neuroinflammation, Neurotransmitters, Neurogenesis, and Synaptic Function. *Front Pharmacol* **11**, 1270, doi:10.3389/fphar.2020.01270 (2020).
- 6 Yu, J. S. *et al.* Lactobacillus lactis and Pediococcus pentosaceus-driven reprogramming of gut microbiome and metabolome ameliorates the progression of non-alcoholic fatty liver disease. *Clin Transl Med* **11**, e634, doi:10.1002/ctm2.634 (2021).
- 7 Liu, Z. *et al.* The modulatory effect of infusions of green tea, oolong tea, and black tea on gut microbiota in high-fat-induced obese mice. *Food Funct* **7**, 4869-4879, doi:10.1039/c6fo01439a (2016).
- 8 Gribble, F. M. & Reimann, F. Function and mechanisms of enteroendocrine cells and gut hormones in metabolism. *Nat Rev Endocrinol* **15**, 226-237, doi:10.1038/s41574-019-0168-8 (2019).
- 9 Brighton, C. A. *et al.* Bile Acids Trigger GLP-1 Release Predominantly by Accessing Basolaterally Located G Protein-Coupled Bile Acid Receptors. *Endocrinology* **156**, 3961-3970, doi:10.1210/en.2015-1321 (2015).
- 10 Trabelsi, M. S. *et al.* Farnesoid X receptor inhibits glucagon-like peptide-1 production by enteroendocrine L cells. *Nat Commun* **6**, 7629, doi:10.1038/ncomms8629 (2015).

Reviewers' comments:

Reviewer #1 (Remarks to the Author):

The authors have replied to all my comments and provided comprehensive results in the revised manuscript.

Several comments:

1. In Fig 1, the authors have updated the postprandial GLP-1. Is this after the OGTT? When did the author measure the postprandial GLP-1 after the meal?

2. The description in line 122 to 125 is incorrect. The authors stated that 'The blood glucose levels at all time points after glucose administration were significantly lower in the HTH group compared to control mice (Supplementary Figure2. i). This finding was further supported by higher Area Under the Curve (AUC) values in the HTH group compared to control mice (Supplementary Figure2. J).'

If the blood glucose levels were significantly lower in the HTH group than the control group, the AUC should be consistent - lower blood glucose AUC in HTH than control group.

3. In relation to the above comment. The authors found that the blood glucose levels are lower in the HTH group, why do the authors think that HTH is a 'bad' thing?

4. Minor comment: what is the rationale of the HTH treatment period? I would suggest the authors provide a reference.

Reviewer #2 (Remarks to the Author):

I would like to thank the authors for their careful responses to my queries. I am happy to see the addition of the food intake data, the new inflammation data and the clarity on FDR correction, and I agree with the removal of the data in Fig 2I-J and in Fig7C-D.

However, the authors removed quite a bit of other data during revisions and although I understand the desire to streamline, I feel the amount of data removed was excessive and its loss hurts the overall story. In particular:

1. I liked the oxygen data and I feel it still belongs in the manuscript alongside the inflammation data. Increased oxygenation of the gut is consistent with the decrease in butyrate, via reduced colonocyte metabolism overall (e.g. see Litkav, Byndloss & Bäumlér (2018) for a review on the subject <https://www.science.org/doi/10.1126/science.aat9076>.) Together, the oxygen and inflammation data provide some mechanistic rationale for the microbiome 'dysbiosis' observed, since the microbiome is full of strict anaerobes and is very sensitive to oxygen levels; in my mind, this therefore makes a more convincing story.

2. The fecal bile acids data originally shown in 4J also seem to me to be an important part of the story since they confirm that changes seen in the liver are also present in the gut where the microbiota 'action' is. I would also re-include this data.

3. I also disagree with the removal of the second microbiome sequencing run in response to my concern about reproducibility. First, because I think it is better to acknowledge variability than to try and hide it (and the microbiome is famously variable between mouse cages, housing institutions and experiments). Second, because the data are still used in a correlation analysis in Fig 7 (right?), and so the original data are important for the reader's understanding of that analysis.

4. Also, I think my original comment #2 about microbiome experimental reproducibility was slightly misunderstood: I wanted to know if all 6 mice in Exp. 1 were housed in the same cage or were spread across 2 cages, and similarly for Exp. 3. Mouse microbiomes are often similar by cage due to coprophagy, so this information is useful for interpretation. I suggest including it in the methods section or in Fig S1.

Additional comments:

- Line 194 – succinic acid is not a bile acid.
- Line 207 – clarify what Ncl-716 cells are (human colonic?)
- Line 344 – this doesn't really make sense, many Firmicutes members synthesize butyrate so you would expect increased Firmicutes to yield increased butyrate, if anything (but it can vary by species). Could just remove this.
- I appreciate the authors' efforts to streamline and clarify the writing of the paper, however it is still relatively dense and difficult to read. One suggestion might be to include a diagram summarizing the main findings at the end of the paper, e.g. the hypothesized link between HCH, microbiome/LCA, antagonism of FXR, and GLP-1, and any other downstream effects.

Reviewer #3 (Remarks to the Author):

The authors are to be commended on the work they have put into reviewing the manuscript and taking on board the comments from the reviewers, in addition to completing additional experimental work to improve this paper. It is improved on the first draft. However, issues still remain given the experimental design is primarily observational - mice are exposed to environmental change and biological readouts are somewhat randomly compared. The research does not indicate how a change in environment modifies the microbiome, GLP-1 secretion or bile acid function.

Detailed Response to Reviewers

We would like to express our sincere gratitude to the editor and reviewers for their thoughtful and constructive feedback on our manuscript, which greatly help us improve the quality of our work. We have carefully addressed all their concerns and suggestions in the revised version of the manuscript. They are described in detail in the following point-to-point session. We hope the manuscript can be considered for publication in its revised version.

Comments from the Reviewers:

Reviewer #1: The authors have replied to all my comments and provided comprehensive results in the revised manuscript.

Reviewer point #1: In Fig 1, the authors have updated the postprandial GLP-1. Is this after the OGTT? When did the author measure the postprandial GLP-1 after the meal?

Author response #1:

Thank you for your questions regarding the details of Fig 1 and the measurement of postprandial GLP-1 levels. Yes, the postprandial GLP-1 levels shown in the figure were measured following an Oral Glucose Tolerance Test (OGTT). Specifically, the OGTT was conducted on the day prior to the sample collection. Then, on the day of sample collection, we measured the postprandial GLP-1 levels ten minutes after meal. We hope this clarifies the methodology used in our study, and we are grateful for your attention to this detail.

Reviewer point #2: The description in line 122 to 125 is incorrect. The authors stated that 'The blood glucose levels at all time points after glucose administration were significantly lower in the HTH group compared to control mice (Supplementary Figure2. i). This finding was further supported by higher Area Under the Curve (AUC) values in the HTH group compared to control mice (Supplementary Figure2. J).' If the blood glucose levels were significantly lower in the HTH group than the control group, the AUC should be consistent - lower blood glucose AUC in HTH than control group.

Author response #2:

Thank you for pointing out this mistake in our manuscript. Upon review, I acknowledge the error in our original statement regarding the blood glucose levels and the Area Under the Curve (AUC) values in the HTH group compared to control mice. The correct information should read as follows: "The blood glucose levels at all time points after glucose administration were significantly lower in the HTH group compared to control mice (Supplementary Figure2. i). This finding was further supported by lower Area Under the Curve (AUC) values in the HTH group compared to control mice (Supplementary Figure2. j)."

We apologize for the oversight and any confusion it may have caused. The manuscript has been revised to reflect this correction, ensuring accuracy and clarity in our findings. We appreciate your meticulous review and assistance in improving the quality of our work.

Figure S2. Phenotypic parameters after high-temperature and humidity (HTH) exposure in BABL/c mice.

(a) Body weight (*mean*±*s.e.m*) (b) Plasma fasting glucose levels. (c) Plasma peptide YY (PYY) levels. (d) Plasma gastric inhibitory polypeptide (GIP) levels. (e) Plasma triglyceride levels. (f) Plasma high-density lipoprotein (HDL) levels. (g) Plasma low-density lipoprotein (LDL) levels. (h) Plasma total cholesterol levels. The difference between HTH and normal control (NC) groups was not significant. (i) and (j) OGTT. (k) The mRNA expression of IL-6, MCP-1, TNF- α in colon. $n=6$ mice for each group. * $P<0.05$, ** $P<0.01$, *** $P<0.001$ by two-tailed Student's t-test. Error bars represent \pm s.d.

Reviewer point #3: In relation to the above comment. The authors found that the blood glucose levels are lower in the HTH group, why do the authors think that HTH is a 'bad' thing?

Author response #3:

Thank you for your query regarding our interpretation of HTH in our study. I apologize for any confusion caused by our description. I would like to clarify that we do not categorize HTH as inherently 'bad' in our research. Our study reveals that HTH is associated with reduced levels of Lithocholic Acid (LCA), which we found to have a suppressive effect on GLP-1 secretion, mediated through the activation of the Farnesoid X Receptor (FXR). This specific pathway and its implications are the core focus of our research, rather than a qualitative assessment of HTH as 'good' or 'bad'.

We appreciate your insightful question, which has highlighted a need for clearer communication in our manuscript. We will ensure that this aspect is more accurately and objectively presented in our revised manuscript.

Reviewer point #4: Minor comment: what is the rationale of the HTH treatment period? I would suggest the authors provide a reference.

Author response #4

Thank you for your insightful comment regarding the rationale of the HTH treatment period in our study. Our decision for the duration of the HTH treatment was referred to our previous method. Specifically, the treatment period is based on experimental designs and findings from our prior studies, which have been published. To provide a reference for this methodology, our previous works titled “Gut Dysbiosis with Minimal Enteritis Induced by High Temperature and Humidity” in the Scientific Reports journal, and “Gut microbiota associated with appetite suppression in high-temperature and high-humidity environments” in EBioMedicine, offer a detailed background on the rationale and implementation of the HTH treatment period.

We have included this information in the **Methods** section of our revised manuscript. We believe this inclusion will aid in understanding the foundation and reasoning behind our approach in this study. Thank you for bringing this to our attention, and we appreciate your suggestion to enhance the manuscript:

Page 11, Line 470-473

Method: The climate chambers were set to temperatures of 32 ± 1 °C and relative humidity (RH) levels of 85-90% (HTH group) or solely 85-90% RH (HH group) over 14 consecutive days as previous described^{1,2}.

Reviewer #2:

I would like to thank the authors for their careful responses to my queries. I am happy to see the addition of the food intake data, the new inflammation data and the clarity on FDR correction, and I agree with the removal of the data in Fig 2I-J and in Fig7C-D.

However, the authors removed quite a bit of other data during revisions and although I understand the desire to streamline, I feel the amount of data removed was excessive and its loss hurts the overall story. In particular:

Reviewer point #1: I liked the oxygen data and I feel it still belongs in the manuscript alongside the inflammation data. Increased oxygenation of the gut is consistent with the decrease in butyrate, via reduced colonocyte metabolism overall (e.g. see Litkav, Byndloss & Bäumler (2018) for a review on the subject <https://www.science.org/doi/10.1126/science.aat9076>).

Together, the oxygen and inflammation data provide some mechanistic rationale for the microbiome 'dysbiosis' observed, since the microbiome is full of strict anaerobes and is very sensitive to oxygen levels; in my mind, this therefore makes a more convincing story.

Author response #1:

Thank you for your valuable feedback regarding the inclusion of oxygen data in our manuscript. We agree with your perspective that the oxygen data, in conjunction with the inflammation data, contributes significantly to the narrative of our study. In light of your suggestion, we have incorporated both oxygen and inflammation data in the revised manuscript.

Page 4, Line 135-144

Results:

Considering the critical role of GLP-1 in energy metabolism and inflammation, we tried to investigate the underlying mechanisms of GLP-1 suppression under HTH environment. As high relative humidity reduces the ambient oxygen concentrations in the air, and hypoxia inhibits GLP-1 production *in vivo*³. We wondered that whether the aggravated degree of gut hypoxia caused the GLP-1 suppression. However, although we observed an augmented staining of Hp-1 (a hypoxia probe indicating the degree of hypoxia) in the liver in HTH group (Fig. 1g and i). Most importantly, out of our expect, the weakened Hp-1 staining in the ileum indicated a significant loss of hypoxia status (Fig. 1h and j). This result indicated a disturbance that liver became hypoxic whereas ileum lost its physiological hypoxia status.

Page 5, Line 186-188

In this study, we also found a significant elevation in the expression levels of IL-6, MCP-1, and TNF- α after 2 weeks of exposure to HTH conditions (Supplementary Figure2. k).

Figure 1. The impact of environmental HH and HTH on bio-clinical parameters in mice.

(a) Plasma levels of fasting GLP-1 (b) ghrelin, (c) insulin, (d) postprandial GLP-1. (e) Food intake amount of Day7. $n = 6$ for each group. * $P < 0.05$, ** $P < 0.01$ by two-tailed Student's t-test. Error bars represent \pm s.d. (f) Food intake amount of Day14. $n = 14$ for each group. * $P < 0.05$ by two-tailed Student's t-test. Error bars represent \pm s.d.

(g, i) The representative images of HP-1 staining of liver. (h, j) The representative images of HP-1 staining of ileum. Scale bars=20 μ m. $n = 6$ for each group. *** $P < 0.001$ by two-tailed Student's t-test. Error bars represent \pm s.d.

Figure S2. Phenotypic parameters after high-temperature and humidity (HTH) exposure in BABL/c mice.

(a) Body weight (*mean* \pm *s.e.m*) (b) Plasma fasting glucose levels. (c) Plasma peptide YY (PYY) levels. (d) Plasma gastric inhibitory polypeptide (GIP) levels. (e) Plasma triglyceride levels. (f) Plasma high-density lipoprotein (HDL) levels. (g) Plasma low-density lipoprotein (LDL) levels. (h) Plasma total cholesterol levels. The difference between HTH and normal control (NC) groups was not significant. (i) and (j) OGTT. (k) The mRNA expression of IL-6, MCP-1, TNF- α in colon. $n=6$ mice for each group. * $P<0.05$, ** $P<0.01$, *** $P<0.001$ by two-tailed Student's t-test. Error bars represent \pm s.d.

Reviewer point #2:

The fecal bile acids data originally shown in 4J also seem to me to be an important part of the story since they confirm that changes seen in the liver are also present in the gut where the microbiota ‘action’ is. I would also re-include this data.

Author response #2:

Thank you for highlighting the importance of the fecal bile acids data in our study. In response to your suggestion, we have re-incorporated the fecal bile acids data into Fig.4h of our revised manuscript. Additionally, we have updated the **Results** section to include a detailed description and analysis of this data.

Page 6, Line 238-241

We also determined the fecal BAs profile. As shown in Fig.4h, similarly as in the liver, we also observed a significant decreased TLCA, as well as the marginally decreased 7-kLCA, 7,12-kLCA, and DHCA, in HTH group.

Figure 4. BAs homeostasis in mice receiving persistent exposure of environmental

HTH.

(a) The diagrammatic sketch of bile acids homeostasis and its regulatory effect on β -oxidation through FXR-ACC2 pathway. The hepatic expression of (b) Cyp7a1, (c) Cyp7b1, (d) Cyp8b1, and (e) Cyp27a1 were shown in columns. n=6 for each group, * P <0.05 by two-tailed Student's t-test. Error bars represent \pm s.d. (f) The level of GLP-1 after treated with LCA, TLCA, CDCA and TCDCA *in vitro*. n=3 for each group, *** P <0.001 by ANOVA and Kruskal-Wallis test. Error bars represent \pm s.d. (g) Targeted metabolomics analysis of BAs in liver. n=10 for each group, * P <0.05 by two-tailed Student's t-test (LCA) and Mann-Whitney test (TLCA). Error bars represent \pm s.d. (h) Targeted metabolomics analysis of BAs in fecal. n=5 for each group, * P <0.05 by two-tailed Student's t-test. Error bars represent \pm s.d.

Reviewer point #3:

I also disagree with the removal of the second microbiome sequencing run in response to my concern about reproducibility. First, because I think it is better to acknowledge variability than to try and hide it (and the microbiome is famously variable between mouse cages, housing institutions and experiments). Second, because the data are still used in a correlation analysis in Fig 7 (right?), and so the original data are important for the reader's understanding of that analysis.

Author response #3:

Thank you for your valuable insights regarding the inclusion of the second microbiome sequencing run in our study. In response to your comments, I have reinstated the results of the second microbiome sequencing run in the revised manuscript. These are now detailed in the **Results** section and visually represented in **Supplementary Figure 3**.

Page 5, Line 201-202

The fecal 16S rRNA gene sequencing was also performed. The genus with differential abundance were shown in Supplementary Figure3.

Figure S3. The relative proportion of genus with differential abundance identified at day 10 in HTH treated mice (Exp.3).

Reviewer point #4:

Also, I think my original comment #2 about microbiome experimental reproducibility was slightly misunderstood: I wanted to know if all 6 mice in Exp. 1 were housed in the same cage or were spread across 2 cages, and similarly for Exp. 3. Mouse microbiomes are often similar by cage due to coprophagy, so this information is useful for interpretation. I suggest including it in the methods section or in Fig S1.

Author response #4:

Thank you for your insightful comments and inquiry regarding the details of our experimental design. Addressing your question about the housing conditions of the mice in Exp 1 and 3, I would like to clarify as follows:

In our study, mice from different groups were housed in separate cages to avoid cross-contamination. Specifically, in Exp 1, each group consisted of 6 mice, and all these mice were housed together in the same cage. Similarly, in Exp 3, each group had 5 mice, and these mice were also housed together in a single cage.

We understand the significance of housing conditions on the microbiome composition,

especially considering the prevalence of coprophagy in mice. We acknowledge that this information is crucial for the interpretation of our results and agree with your suggestion to include these details in the **methods** section for clearer understanding and reproducibility of our experiments. Thank you again for bringing this to our attention, and we will ensure that this information is incorporated in the revised manuscript.

Page 11, Line 474-478

In our study, mice from different groups were housed in separate cages to avoid cross-contamination. Specifically, in Exp 1, each group consisted of 6 mice, and all these mice were housed together in the same cage. Similarly, in Exp 3, each group had 5 mice, and these mice were also housed together in a single cage.

Additional comments:

- ***Line 194 – succinic acid is not a bile acid.***

Author response:

Thank you for pointing out the error in line 194 regarding the misclassification of succinic acid as a bile acid. We acknowledge this mistake and appreciate your attention to detail. In response to your comment, we have removed the incorrect statement from the revised manuscript. We are grateful for your assistance in enhancing the accuracy and quality of our manuscript.

- ***Line 207 – clarify what Ncl-716 cells are (human colonic?)***

Author response:

Thank you for your request for clarification regarding the Ncl-H716 cells mentioned in line 207 of our manuscript. To address this, we have revised the description to provide a clearer understanding of these cells. The revised text now reads: "Cultured Ncl-H716 cells, which are human L-cells, were treated with chenodeoxycholic acid (CDCA), taurochenodeoxycholic acid (TCDCa), and tauroolithocholic acid (TLCA) at concentrations ranging from 25 to 100 μ M. These treatments acted as agonists and exhibited a dose-dependent effect on GLP-1 production."(page6: line228-230)

We hope this revision adequately addresses your query and enhances the clarity of our manuscript.

- ***Line 344 – this doesn't really make sense, many Firmicutes members synthesize***

butyrate so you would expect increased Firmicutes to yield increased butyrate, if anything (but it can vary by species). Could just remove this.

Author response:

Thank you for your observation regarding the statement in line 344 about the relationship between *Firmicutes* and butyrate production. In light of your feedback, we have removed the aforementioned statement from the revised manuscript. We appreciate your guidance in ensuring the scientific accuracy and clarity of our manuscript.

• I appreciate the authors' efforts to streamline and clarify the writing of the paper, however it is still relatively dense and difficult to read. One suggestion might be to include a diagram summarizing the main findings at the end of the paper, e.g. the hypothesized link between HCH, microbiome/LCA, antagonism of FXR, and GLP-1, and any other downstream effects.

Author response:

Thank you for your valuable suggestion to include a diagram that summarizes the main findings of our paper. We appreciate your feedback regarding the complexity of the text, and agree that a visual representation could greatly enhance the clarity and comprehensibility of our findings.

In response to your suggestion, we have created a comprehensive diagram that succinctly illustrates the hypothesized links explored in our study. This diagram graphically illustrates that exposure to HTH disrupts the gut microbiome, leading to a reduction in LCA levels. This decrease in LCA triggers an upregulation of FXR expression, ultimately resulting in diminished GLP-1 secretion. The diagram has been placed at the end of the manuscript for easy reference and to serve as a summary of our key findings.

Thank you again for your constructive comments, which have improved the quality of our manuscript.

Diagram

Reviewer #3:

The authors are to be commended on the work they have put into reviewing the manuscript and taking on board the comments from the reviewers, in addition to completing additional experimental work to improve this paper. It is improved on the first draft. However, issues still remain given the experimental design is primarily observational - mice are exposed to environmental change and biological readouts are somewhat randomly compared.

The research does not indicate how a change in environment modifies the microbiome, GLP-1 secretion or bile acid function

Author response:

Thank you for acknowledging the efforts we have put into revising the manuscript and for your constructive feedback, which has been instrumental in enhancing the quality of our work.

Firstly, we agree that while our study demonstrates that HTH can alter gut microbiota and bile acids, it does not explicitly unravel the mechanistic pathways of these changes. We acknowledge this as a limitation in our current research. And we have added the

conclusion and limitation in our revised manuscript.

Page10-11, line444-449

“In summary, we demonstrated that exposure to HTH disrupts the gut microbiome, leading to a reduction in LCA levels. This decrease in LCA triggers an upregulation of FXR expression, ultimately resulting in diminished GLP-1 secretion (shown in the diagram). However, our research does not explicitly unravel the mechanistic pathways which govern the changes of the gut microbiome and BAs upon HTH. In future research, we aim to conduct more focused experiments to elucidate these direct mechanisms.”

Once again, we are grateful for your insights and guidance, which have been invaluable in refining our study.

Diagram

- 1 Wu, Y. *et al.* Gut microbiota associated with appetite suppression in high-temperature and high-humidity environments. *EBioMedicine* **99**, 104918, doi:10.1016/j.ebiom.2023.104918 (2024).
- 2 Chen, S. *et al.* Gut Dysbiosis with Minimal Enteritis Induced by High Temperature and Humidity. *Sci Rep* **9**, 18686, doi:10.1038/s41598-019-55337-x (2019).

- 3 Duraisamy, A. J. *et al.* Changes in ghrelin, CCK, GLP-1, and peroxisome proliferator-activated receptors in a hypoxia-induced anorexia rat model. *Endokrynologia Polska* **66**, 334-341, doi:10.5603/EP.2015.0043 (2015).

I would like to extend my heartfelt thanks for providing me with the opportunity to revise our manuscript. Your detailed and insightful comments have been instrumental in enhancing the overall quality and rigor of our work. The thoroughness of your reviews has not only helped in identifying areas that needed improvement but also in guiding our thought process towards a more comprehensive and clearer presentation of our research.

The opportunity to address your concerns and suggestions has been invaluable. It has allowed us to refine our arguments and clarify our conclusions, thereby contributing significantly to the advancement of our study. I am grateful for the time and effort you have dedicated to reviewing our work, and for the constructive criticism that has pushed us to strive for excellence.

We hope that the revisions we have made are in line with your expectations.

Thank you once again for your invaluable input and for the chance to improve our manuscript.

REVIEWERS' COMMENTS:

Reviewer #1 (Remarks to the Author):

The authors have addressed my comments. I have no further questions.

Reviewer #2 (Remarks to the Author):

The authors have responded satisfactorily to my concerns.

Editor remark: Mentions to us editors that text is very dense, sometimes unclear and several figures are not appealing. We do have a lot of notes specifically for this and also recommend a scientific language editing service.